# Opioid-related treatment, interventions, and outcomes among incarcerated persons: A systematic review

Monica Malta[1,2,3]*, Thepikaa Varatharajan[1], Cayley Russell[1], Michelle Pang[1], Sarah Bonato[4], Benedikt Fischer[5,6,7]

1 Institute for Mental Health Policy Research, Centre for Addiction and Mental Health, Toronto, Ontario, Canada, 2 Department of Psychiatry, University of Toronto, Toronto, Ontario, Canada, 3 Social Science Department, Sergio Arouca National School of Public Health, Oswaldo Cruz Foundation, Rio de Janeiro, Brazil, 4 Office of Education, Centre for Addiction and Mental Health, Toronto, Ontario, Canada, 5 Centre for Applied Research in Mental Health and Addiction, Faculty of Health Sciences, Simon Fraser University, Vancouver, British Columbia, Canada, 6 Department of Psychiatry, Federal University of São Paulo, São Paulo, Brazil, 7 Faculty of Medical and Health Sciences, University of Auckland, Auckland, New Zealand

* monica.malta@camh.ca

## Abstract

### Background

Worldwide opioid-related overdose has become a major public health crisis. People with opioid use disorder (OUD) are overrepresented in the criminal justice system and at higher risk for opioid-related mortality. However, correctional facilities frequently adopt an abstinence-only approach, seldom offering the gold standard opioid agonist treatment (OAT) to incarcerated persons with OUD. In an attempt to inform adequate management of OUD among incarcerated persons, we conducted a systematic review of opioid-related interventions delivered before, during, and after incarceration.

### Methods and findings

We systematically reviewed 8 electronic databases for original, peer-reviewed literature published between January 2008 and October 2019. Our review included studies conducted among adult participants with OUD who were incarcerated or recently released into the community (≤90 days post-incarceration). The search identified 2,356 articles, 46 of which met the inclusion criteria based on assessments by 2 independent reviewers. Thirty studies were conducted in North America, 9 in Europe, and 7 in Asia/Oceania. The systematic review included 22 randomized control trials (RCTs), 3 non-randomized clinical trials, and 21 observational studies. Eight observational studies utilized administrative data and included large sample sizes (median of 10,419 [range 2273–131,472] participants), and 13 observational studies utilized primary data, with a median of 140 (range 27–960) participants. RCTs and non-randomized clinical trials included a median of 198 (range 15–1,557) and 44 (range 27–382) participants, respectively. Twelve studies included only men, 1 study included only women, and in the remaining 33 studies, the percentage of women was below 30%. The majority of study participants were middle-aged adults (36–55 years). Participants

**Funding:** BF acknowledges funding from the Canadian Institutes of Health Research (CIHR) for the Ontario CRISM Node Team (grant #SMN-139150); support from the Chair in Addiction, Department of Psychiatry at the University of Toronto for partial development of the article; and support from the Hugh Green Foundation Chair in Addiction Research, Faculty of Medical and Health Sciences at the University of Auckland. MM acknowledges funding from Correctional Service of Canada - CSC (MOU #15-070). The funders had no role in study design, data collection and analysis, decision to publish, or preparation of the manuscript.

**Competing interests:** The authors have declared that no competing interests exist.

**Abbreviations:** AHR, adjusted hazard ratio; AOR, adjusted odds ratio; BMT, buprenorphine maintenance treatment; BPN, buprenorphine; HR, hazard ratio; IRR, incidence rate ratio; MMT, methadone maintenance treatment; MR, mortality rate; NLX, naloxone; NNP, National Naloxone Program; OAT, opioid agonist treatment; OUD, opioid use disorder; PY, person-years; RCT, randomized control trial; RIDOC, Rhode Island Department of Corrections; XR-NTX, injectable extended-release naltrexone.

treated at a correctional facility with methadone maintenance treatment (MMT) or buprenorphine (BPN)/naloxone (NLX) had lower rates of illicit opioid use, had higher adherence to OUD treatment, were less likely to be re-incarcerated, and were more likely to be working 1 year post-incarceration. Participants who received MMT or BPN/NLX while incarcerated had fewer nonfatal overdoses and lower mortality. The main limitation of our systematic review is the high heterogeneity of studies (different designs, settings, populations, treatments, and outcomes), precluding a meta-analysis. Other study limitations include the insufficient data about incarcerated women with OUD, and the lack of information about incarcerated populations with OUD who are not included in published research.

## Conclusions

In this carefully conducted systematic review, we found that correctional facilities should scale up OAT among incarcerated persons with OUD. The strategy is likely to decrease opioid-related overdose and mortality, reduce opioid use and other risky behaviors during and after incarceration, and improve retention in addiction treatment after prison release. Immediate OAT after prison release and additional preventive strategies such as the distribution of NLX kits to at-risk individuals upon release greatly decrease the occurrence of opioid-related overdose and mortality. In an effort to mitigate the impact of the opioid-related overdose crisis, it is crucial to scale up OAT and opioid-related overdose prevention strategies (e.g., NLX) within a continuum of treatment before, during, and after incarceration.

---

### Author summary

#### Why was this study done?

- Opioid use disorder has been rising at an alarming rate, and opioid-related overdose is now a major public health crisis.

- Persons with opioid use disorder are overrepresented in the criminal justice system and face higher risks for opioid-related mortality.

- However, opioid use treatment is severely limited in correctional facilities.

- To address the opioid-related overdose crisis, it is pivotal to improve access to opioid use treatment inside correctional facilities and to assure proper linkage into addiction care post-incarceration.

#### What did the researchers do and find?

- We conducted a systematic review of opioid-related interventions delivered before, during, and after incarceration.

- Our search identified 2,356 scientific articles, of which 46 studies were eligible for inclusion in our review.

- Participants treated at a correctional institution with the gold standard treatment for opioid use disorders, opioid agonist treatment, had higher adherence to addiction

treatment, had lower rates of relapse into illicit opioid use, were less likely to be re-incarcerated, and were more likely to be working 1 year post-incarceration.

- Participants who received opioid agonist treatment while incarcerated and were adequately linked into care post-release experienced a significant decrease in nonfatal overdose rates and mortality.

### What do these findings mean?

- In an effort to mitigate the impact of the opioid-related overdose crisis, it is crucial to scale up opioid-related treatment and prevention strategies within a continuum of treatment before, during, and after incarceration.

## Introduction

Globally the rate of opioid use disorder (OUD) has been rising at an alarming rate, and the opioid-related overdose crisis is now considered a major global challenge, associated with high rates of morbidity and mortality [1]. Unprecedented levels of opioid use and misuse and related health harms (e.g., hospitalizations, treatment admissions, overdose fatalities) have been reported worldwide in recent years, but have been most extensively documented in North America [1–4].

The pervasive general population trends of OUD and the opioid-related overdose crisis disproportionately affect criminal-justice-involved populations [5,6]. It is estimated that more than one-fourth of people with OUD in the United States pass through prisons and jails every year [7]. Worldwide, the United Nations Office on Drugs and Crime reports that 10% of incarcerated individuals have used heroin at some point during incarceration, with one-third reporting current (i.e., past month) use while incarcerated [8]. Similarly, individuals who used opioids prior to their incarceration often report continued use during the course of their incarceration [9]. A report from the US Department of Justice indicated that the percentage of US-state-incarcerated individuals with a diagnosis of drug dependence or abuse is 14 times higher than that of the general population (4%) [9]. A meta-analysis conducted by Fazel et al. [10] estimated a pooled prevalence of 30% and 51% for substance use disorders among incarcerated males and females, respectively, and revealed an overall increasing trend in the prevalence of substance use disorders among incarcerated individuals during recent decades.

Problematic substance use is associated with a range of harmful effects, including the likelihood of continued involvement with the criminal justice system [11] related to the societal responses to substance use, often centered on punishment and mass incarceration [12] rather than public health interventions, including harm reduction strategies and opioid agonist treatment (OAT) [13]. Adverse health effects are also exacerbated when individuals have little control over their environment, such as in a correctional institution [14]. Opioid-related overdose is a leading cause of death among the correctional population, during or after incarceration [5,6,15]. In the first few weeks after release from incarceration, individuals are at an increased risk of opioid use relapse and opioid-related overdose, likely higher when riskier patterns of substance use are adopted and in the absence of proper OAT or with treatment discontinuation [5,6,16].

Furthermore, incarcerated individuals who use opioids may further engage in high-risk behaviors such as needle sharing and/or reutilization, thus increasing the risk of acquiring blood-borne infections (e.g., HIV, hepatitis C) [1,7,17,18]. These factors compound and contribute to incarcerated individuals' disproportionately worse health status relative to the general population [10,12].

Among the general population with diagnosed OUD, OAT is highly effective [19]. OAT decreases the rates of opioid use relapse and transmission of HIV and other blood-borne infections, reduces opioid-related and all-cause mortality [20], and is associated with a broad range of personal and social gains, such as improvements in employment rates and better family functioning [14].

In spite of the disproportional burden of OUD among incarcerated populations, and the well-known benefits of OAT, interventions to address OUD-related risks and harms among this population have been limited [21]. Few criminal justice facilities routinely and adequately screen their populations for OUD, and a smaller percentage provide OAT for incarcerated individuals with diagnosed OUD [12,13]. Significant barriers to adopting and routinizing OAT in correctional institutions exist, and concerns range from medication diversion and safety to constraints related to organizational resources, prohibitive legislations, and continuity of care [22,23].

Several potential linkage points throughout the criminal justice continuum exist, providing opportunities to screen individuals for OUD and overdose risk, and to provide prevention and treatment interventions (e.g., during law enforcement interactions, court hearings, incarceration, community re-entry, and community supervision, such as parole and probation) [11]. Interventions can also be adapted to each step of the continuum, reflecting the different capabilities and programming needs of different correctional institutions [13]. As such, correctional institutions present a unique opportunity to provide prevention, treatment, and post-release support for incarcerated individuals with OUD, thereby encouraging positive health and community reintegration outcomes post-release [12,13,24,25].

As problematic opioid use and its related harms continue to inflict a major public health burden on many regions, especially North America [1]—and disproportionately so among correctional populations [26]—a comprehensive and timely examination of the extant international literature is warranted in order to inform effective program and policy measures that sufficiently address the complex needs of incarcerated opioid-involved individuals [12,13]. The objective of this systematic review was to assess opioid-related interventions delivered during and after incarceration among adult correctional populations. Previous reviews have assessed the effectiveness of various opioid-related intervention modalities among correctional populations [25,27,28]. However, the present systematic review is the first to our knowledge to systematically review the literature to assess the effects of both treatment-based (e.g., methadone, buprenorphine [BPN], naltrexone) and preventive (e.g., naloxone [NLX]) opioid-related interventions delivered during and after incarceration among adult correctional populations. The study addresses the impact of the opioid-related overdose crisis and highlights effective public health strategies developed and evaluated among a highly disenfranchised and frequently forgotten population: incarcerated persons with OUD.

## Methods

### Search strategy

We identified all peer-reviewed original research articles in which opioid use interventions delivered to adult correctional populations were evaluated. The search strategy was developed by an experienced health science librarian (SB), and included studies published between

January 1, 2008, and December 31, 2018. After initial peer review, the search was updated to October 17, 2019. The search did not include studies published before 2008 to avoid replication of findings from previous systematic reviews with similar objectives [25,27–31].

This present systematic review adheres to the guidelines outlined by the Cochrane Collaboration [32]. Results are reported according to the PRISMA 2009 checklist [33] (S1 PRISMA Checklist). The review was submitted to PROSPERO (S1 Prospero Registration), an international prospective registry of systematic reviews (registration #135900). The initial search strategy was developed in MEDLINE and combined relevant MeSH terms and keywords regarding opioid use interventions (e.g., opiate substitution treatment, buprenorphine, methadone, naltrexone, naloxone) among incarcerated and post-incarceration populations (i.e., jail, prison, offender, detention, imprison, post-incarceration) that use opioids (i.e., opioid-related disorders, opiate, opioid). The complete MEDLINE search strategy is described on S1 MEDLINE Search Strategy. Detailed search strategies were subsequently developed and revised for each individual database, based on the initial MEDLINE search strategy. Reference lists of all articles selected for inclusion in the review were also scanned for additional relevant studies. The search included the following scientific literature databases: Criminal Justice Abstracts, Embase, MEDLINE, National Criminal Justice Reference Service (NCJRS), PsycINFO, Scopus, and Web of Science. Bibliographic references were managed using the software EndNote X9.

## Inclusion/Exclusion criteria

The review included studies conducted among adult participants who (1) were opioid users at the time of the study and/or had been diagnosed with OUD prior to or during incarceration and (2) were incarcerated or recently released into the community (≤90 days post-incarceration). Studies were excluded if participants were (1) not opioid users, (2) using opioids for medical purposes (not including for OUD), (3) released from incarceration for more than 90 days, (4) on probation or parole at the time of the study, or (5) involved in drug treatment court or other diversion programs. Only studies involving opioid use interventions (i.e., addiction treatment, relapse, or overdose prevention) for adults provided during incarceration or within 90 days post-incarceration were included for final review. Exclusion criteria were developed to select studies directly addressing the impact of providing OAT or opioid-overdose prevention (e.g., NLX) to incarcerated or recently released persons. Studies reporting the following intervention outcomes were included: mortality (e.g., all cause, opioid related), outcomes related to opioid addiction treatment (e.g., relapse, retention, withdrawal), and recidivism (e.g., re-incarceration, arrest). Studies presenting only cost-effectiveness analyses, opioid treatment impact on blood-borne and/or sexually transmitted infections, or self-screening or individual's subjective experiences/feelings/attitudes towards opioid treatment were excluded.

## Data extraction and analysis

Two reviewers (CR and TV) independently screened all articles in a 2-step screening process—first screening the titles/abstracts followed by the full-text articles. When consensus could not be reached among reviewers, a third reviewer (MM) became involved to resolve standing conflicts.

Relevant information was extracted and inputted into a standardized form, which included the following information: year of publication, location and setting of study, sample size, population characteristics (age, sex, and ethnicity), study design, intervention description, and study outcomes. All eligible studies were assessed for quality using the Joanna Briggs Institute Critical Appraisal Tools [34].

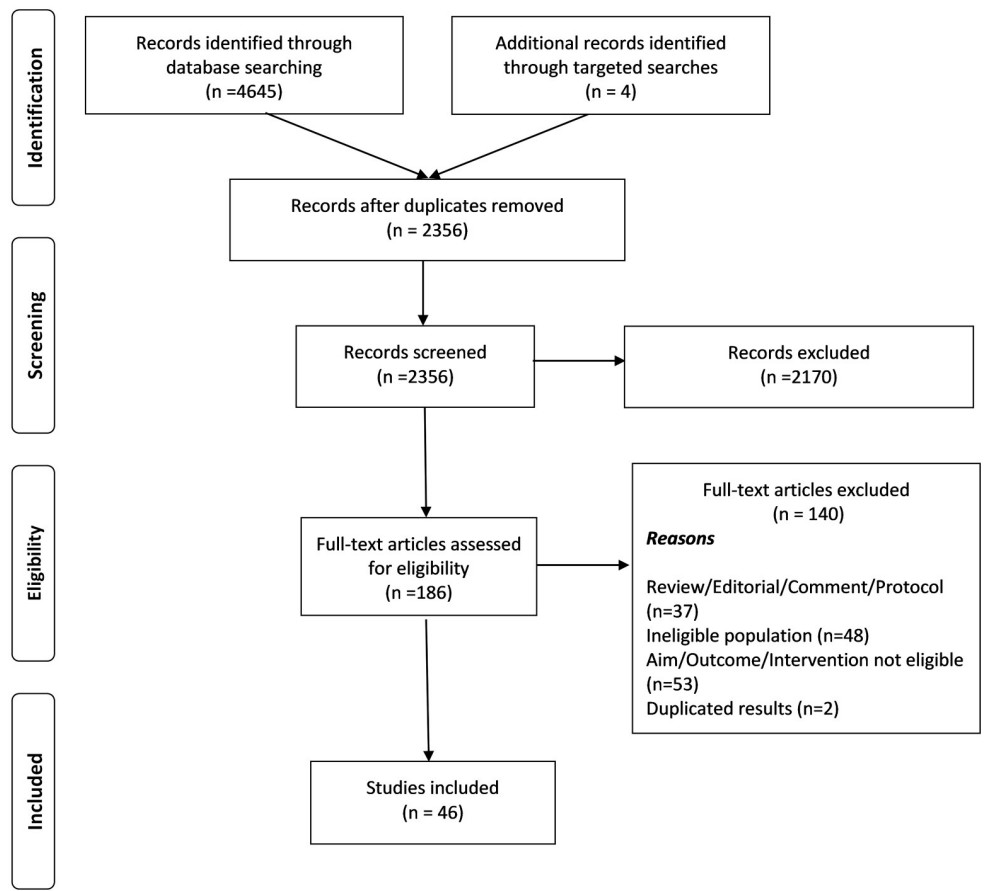

**Fig 1. Flow diagram of search strategy (2008–2019).**

## Results

From the initial search, 2,356 unique peer-reviewed articles were identified. Of these, there was perfect agreement between reviewers on the eligibility of 186 articles, and the ineligibility of 2,170 abstracts that failed to meet the study inclusion criteria. At second screening, a total of 140 articles (S1 List of Manuscripts Excluded) were excluded for the following reasons: (1) were reviews, editorials, comments, or study protocols; (2) studied non-incarcerated individuals with OUD; (3) did not evaluate an opioid use intervention delivered during incarceration or within 90 days post-incarceration; and/or (4) reported duplicate results. Agreement on full-text screening was close to perfect. Ultimately, 46 original articles were deemed eligible for data extraction (Fig 1).

### Study characteristics

Characteristics of the 46 identified studies are summarized in Table 1. While the search included articles published in any language, all identified studies were published in English, and nearly all were conducted in high-income countries. Thirty studies were conducted in North America (US, 27; Canada, 3), 9 were conducted in Europe (United Kingdom, 7; France, 1; Norway, 1), and 7 were conducted in Asia/Oceania (Australia, 4; Malaysia, 2; Taiwan, 1).

Eight studies utilized administrative data and included large sample sizes (median of 10,419 [range 2,273–131,472] participants) [35,36,53,54,68,69,74,75]. All other studies assessed

**Table 1. Characteristics of included studies, 2008–2019.**

| Source | Study location | N | Design (period) | Setting | Characteristics of study population | | | | |
|---|---|---|---|---|---|---|---|---|---|
| | | | | | Population | Age (years)* | Sex | Ethnicity (%) | Addiction criteria |
| Green et al., 2018 [26] | RI, US | 35 | Cohort (2016–2017) | Prison | Individuals with recent incarceration (less than 12 months) who died from overdose | Deaths recorded in 2016: 18–29 y (30.8%), 30–39 y (34.6%), 40–49 y (23.1%), ≥50 y (11.5%); deaths recorded in 2017: 18–29 y (22.2%), 30–39 y (44.4%), 40–49 y (33.3%), ≥50 y (—) | Deaths recorded in 2016: male, 92.3%; female, 7.7%; deaths recorded in 2017: male, 77.8%; female, 22.2% | Deaths recorded in 2016: white, 96.2%; other, 3.8%; deaths recorded in 2017: white, 88.9%; other, 11.1% | Death recorded among individuals who received OAT in prison |
| Degenhardt et al., 2014 [35] | Australia | 16,453 | Cohort (2000–2012) administrative data | Prison | Incarcerated heroin users from prisons in New South Wales | Age at first criminal charge: 23 y (10–64) | Male, 78.7%; female, 21.3% | Indigenous, 29.9% | Incarcerated persons who received MMT/BPN prior incarceration |
| Larney et al., 2014 [36] | Australia | 16,715 | Cohort (2000–2012) administrative data | Prison | Incarcerated heroin users from prisons in New South Wales | Age at first entry into prison: 30 y (16–64) | Male, 78.9%; female, 21.1% | Indigenous, 29.9% | Incarcerated persons who received MMT/BPN prior incarceration |
| Gordon et al., 2014 [37] | MD, US | 211 | RCT (2008–2012) | Prison | Incarcerated persons in pre-release prison | 39.1 y ± 8.8 | Male, 70.1%; female, 29.9% | African-American, 70.1%; white, 25.6%; other, 4.3% | DSM-IV for opioid dependence |
| Gordon et al., 2012 [38] | MD, US | 211 | RCT (2003–2005) | Prison | Men incarcerated at a Baltimore pre-release facility | 35–45 y | 100% male | Majority African-Americans, >60% | DSM-IV for opioid dependence |
| Gordon et al., 2008 [39] | MD, US | 201 | RCT (2003–2005) | Prison | Men incarcerated at a Baltimore pre-release facility | 35–45 y | 100% male | Majority African-Americans, >60% | DSM-IV for opioid dependence |
| Gordon et al., 2017 [40] | MD, US | 211 | RCT (2008–2012) | Prison | Incarcerated persons in pre-release prison | 39.1 y ± 8.8 | Male, 70.1%; female, 29.9% | African-American, 70.1%; white, 25.6%; other, 4.3% | DSM-IV for opioid dependence |
| Gordon et al., 2015 [41] | MD, US | 27 | Clinical trial (2012–2014) | Prison | Incarcerated persons eligible for release within 30 days from screening | 39.9 y ± 8.3 | Male, 59.3%; female, 40.7% | African-American, 85.2%; white, 14.8%; other, 8.3% | DSM-IV for opioid dependence |
| Gordon et al., 2018 [42] | MD, US | 199 | RCT (2008–2012) | Prison | Incarcerated persons in pre-release prison | 39.4 y ± 8.5 | Male, 70.9%; female, 29.1% | African-American, 69.8%; white, 25.6%; other, 4.5% | DSM-IV for opioid dependence |
| Kinlock et al., 2008 [43] | MD, US | 197 | RCT (2003–2005) | Prison | Men incarcerated at a Baltimore pre-release facility | Counseling only, 40.8 y ± 7.7; counseling + transfer to MMT post-release, 40.3 y ± 7.0; counseling + MMT in prison and transfer, 39.8 y ± 7.1 | 100% male | Counseling only: African-American, 65.1%; white, 31.7%; other, 3.2%; counseling + transfer to MMT post-release: African-American, 72.7%; white, 19.7%; other, 7.6%; counseling + MMT in prison and transfer: African-American, 70.6%; white, 20.6%; other, 8.8% | DSM-IV for opioid dependence |

(*Continued*)

**Table 1.** (Continued)

| Source | Study location | N | Design (period) | Setting | Characteristics of study population | | | | |
|--------|----------------|---|-----------------|---------|-----------|-------------|-----|---------------|------------------|
| | | | | | Population | Age (years)* | Sex | Ethnicity (%) | Addiction criteria |
| Kinlock et al., 2009 [44] | MD, US | 204 | RCT (2003–2005) | Prison | Men incarcerated at a Baltimore pre-release facility | 40.3 y ± 7.1 | 100% male | African-American, 69.6%; white, 24.0%; other, 6.4% | DSM-IV for opioid dependence |
| Kinlock et al., 2013 [45] | MD, US | 67 | RCT (2003–2005) | Prison | Men incarcerated at a Baltimore pre-release facility who received MMT in prison | 39.8 y ± 7.1 | 100% male | African-American, 70.6%; white, 20.6%; other, 8.8% | DSM-IV for opioid dependence |
| McKenzie et al., 2012 [46] | RI, US | 62 | RCT (2006–2009) | Prison and jail | Incarcerated persons from RIDOC | 40.7 y (22–58) | Male, 70.9%; female, 29.1% | Hispanic/Latino, 21.0%; other, 79.0% | Self-reported heroin injection or enrolled in MMT in the month preceding incarceration |
| Zaller et al., 2013 [47] | RI, US | 44 | Clinical trial (2006–2009) | Prison and jail | Incarcerated persons from RIDOC | 37.3 y ± 7.3 | Male, 84.1%; female, 15.9% | Hispanic/Latino, 29.5%; black/African-American, 2.2%; white, 68.1% | DSM-IV for opioid dependence |
| Lee et al., 2016 [48] | MD, NY, PA, RI, US | 308 | RCT (2009–2013) | Prison and jail | Participants with OUD with recent incarceration (less than 12 months) | XR-NTX, 44.4 y ± 9.2; TAU, 43.2 y ± 9.4 | XR-NTX: male, 84.3%; female, 15.7%; TAU: male, 85.2%; female, 14.8% | XR-NTX: white, 20.4%; black, 53.3%; Hispanic, 24.3%; TAU: white, 19.4%; black, 47.7%; Hispanic, 29.0% | Clinical diagnosis of OUD |
| Friedmann et al., 2018 [49] | RI, US | 15 | RCT (2012–2014) | Prison | Incarcerated persons scheduled to be released within 1–2 months | Pre-release XR-NTX group, 38.9 y; post-release XR-NTX group, 33.6 y | Pre-release XR-NTX group: male, 88.9%; female, 11.1%; post-release XR-NTX group: 100% male | NA | DSM-IV for opioid dependence |
| Soares et al., 2018 [50] | MD, NY, PA, RI, US | 297 | RCT (2009–2013) | Prison and jail | Participants with OUD with recent incarceration (less than 12 months) | NA | XR-NTX: male, 84.3%; female, 15.7%; TAU: male, 85.2%; female, 14.8% | XR-NTX: white, 50.5%; TAU: white, 19.6% | Clinical diagnosis of OUD |
| Larney et al., 2012 [51] | Australia | 375 | Cohort (1997–2006) | Prison | Male heroin users from prisons in New South Wales | 26 y (18–46) | 100% male | Aboriginal or Torres Strait Islander: 24% | Self-reported heroin use/abuse |
| Farrell-MacDonald et al., 2014 [52] | Canada | 137 | Cohort (2003–2008) | Prison | Incarcerated persons with problematic opioid use | MMT in prison but discontinued post-release, 35.4 y ± 8.0; MMT in prison and continued post-release, 33.0 y ± 7.2; non-MMT treated, 31.3 y ± 7.4 | 100% female | MMT discontinued post-release and Aboriginal ancestry, 22.0%; MMT continued post-release and Aboriginal ancestry, 40.0%; non-MMT treated and Aboriginal ancestry, 38.0% | MMT prior to incarceration |
| Larney et al., 2016 [53] | Australia | 8,577 | Cohort (2007–2013) administrative data | Prison | Incarcerated heroin users from prisons in New South Wales | Age at first recorded OST: 32 y (26–38) | Male, 81.0%; female, 19.0% | Indigenous, 14.5% | Incarcerated persons who received MMT/BPN prior incarceration |

(*Continued*)

**Table 1.** (Continued)

| Source | Study location | N | Design (period) | Setting | Characteristics of study population | | | | |
|--------|----------------|---|-----------------|---------|-------------------------------------|--|--|--|--|
| | | | | | Population | Age (years)* | Sex | Ethnicity (%) | Addiction criteria |
| Bird et al., 2015 [54] | Scotland | 131,427 | Cohort (1996–2007) administrative data | Prison | Incarcerated persons released between 1996 and 2007 | NA | Male, 93%; female, 7% | NA | NA |
| Lincoln et al., 2018 [55] | MA, US | 67 | Cohort (2013–2014) | Jail | Incarcerated persons scheduled to be released | XR-NTX (prior to release), 32.9 y (22–60); XR-NTX planned after release, 34.6 y (21–54) | XR-NTX, prior to release: male, 89.4%; female, 10.6%; XR-NTX planned after release: male, 90.0%; female, 10.0% | XR-NTX, prior to release: black/African-American, 6.4%; Hispanic/Latino, 25.5%; white, 68.1%; XR-NTX planned after release: black/African-American, 0%; Hispanic/Latino, 40.0%; white, 60.0% | Clinical diagnosis of OUD + urine test |
| Sheard et al., 2009 [56] | England | 90 | RCT (2004–2005) | Prison | Incarcerated persons from Her Majesty's Prison Leeds | 29.8 y (19–53) | 100% male | NA | History of opiate use, confirmed by urine test |
| Magura et al., 2009 [57] | NY, US | 116 | RCT (2006–2007) | Jail | Heroin-dependent men not enrolled in community methadone treatment and sentenced to 10–90 days in jail | MMT group, 40.7 y ± 9.1; BPN group, 38.4 y ± 7.9 | 100% male | MMT group: black, 25%; Hispanic, 65%; BPN group: black, 25%; Hispanic, 62% | Clinical diagnosis of opioid dependence |
| Awgu et al., 2010 [58] | NY, US | 114 | RCT (2006–2007) | Jail | Heroin-dependent men not enrolled in community methadone treatment and sentenced to less than 1 year in jail | MMT group, 40.8 y ± 9.2; BPN group, 38.4 y ± 7.9 | 100% male | MMT group: black, 25%; Hispanic, 65%; BPN group: black, 26%; Hispanic, 61% | Clinical diagnosis of opioid dependence |
| Wright et al., 2011 [59] | England | 289 | RCT (2006–2008) | Prison | Population from 3 prison healthcare departments | 30.8 y (26.9–34.9) | NA | Methadone: white British, 89.9%; Asian, 2.7%; black, 4.1%; mixed race, 0.7%; white other, 2.7%; BPN: white British, 93.6%; Asian, 2.8%; black, 0.7%; mixed race, 0.7%; white other, 0.7% | History of opiate use, confirmed by urine test |
| Rich et al., 2015 [60] | RI, US | 283 | RCT (2011–2013) | Prison and jail | Incarcerated persons at RIDOC enrolled in MMT at the time of incarceration | 34 y ± 8.4 | Male, 78%; female, 22% | White, 81%; black, 4%; other, 15% | Opioid users under MMT at incarceration |
| Brinkley-Rubinstein et al., 2018 [61] | RI, US | 179 | RCT (2011–2013) | Prison and jail | Incarcerated persons at RIDOC enrolled in MMT at the time of incarceration | 32.6 y (28.4–40.9) | Male, 78.2%; female, 21.8% | White, 78.8%; black, 4.5%; other, 16.8% | Opioid users under MMT at incarceration |

(*Continued*)

**Table 1.** (Continued)

| Source | Study location | N | Design (period) | Setting | Characteristics of study population | | | | |
|--------|----------------|---|-----------------|---------|-------------|-------------|-----|---------------|--------------------|
| | | | | | Population | Age (years)* | Sex | Ethnicity (%) | Addiction criteria |
| Moore et al., 2018 [62] | CT, US | 382 | Clinical trial (2013–2015) | Prison and jail | Incarcerated persons who received MMT 5 days prior to incarceration | MMT in prison, 36.2 y ± 9.6; forced withdrawal, 37.0 y ± 9.1 | 100% male | MMT in prison: white, 78.8%; black, 9.8%; Hispanic, 11.4%; forced withdrawal: white, 76.8%; black, 9.1%; Hispanic, 13.6%; Native American, 0.5% | DSM-IV for opioid dependence |
| McMillan et al., 2008 [63] | US | 589 | Cohort (2005–2006) | Jail | Incarcerated persons released between 11/2005 and 10/2006 | Jail-based MMT, 38.5 y ± 10.0; no MMT, 37.7 y ± 9.9 | Jail-based MMT: 71.5% male; no MMT: 67.5% male | Jail-based MMT: Hispanic, 78.0%; non-Hispanic white, 20.9%; Native American, 1.0%; no MMT: Hispanic, 80.4%; non-Hispanic white, 16.8%; Native American, 2.8% | Incarcerated persons who reported MMT prior to incarceration |
| Marzo et al., 2009 [64] | France | 507 | Cohort (2003–2006) | Prison | Opioid-dependent patients included within the first week of imprisonment | 30.8 y ± 6.4 | Male, 96.3%; female, 3.7% | NA | Clinical evaluation and self-report |
| Wickersham et al., 2013 [65] | Malaysia | 27 | Cohort (2009–2010) | Prison | HIV-positive incarcerated persons (up to 4 months pre-release) | 37.1 y ± 7.1 | 100% male | Malay, 73.3%; Indian, 20.0%; Chinese, 6.7% | DSM-IV for opioid dependence |
| Wickersham et al., 2013 [66] | Malaysia | 72 | Cohort (2008–2009) | Prison | Incarcerated persons receiving MMT and scheduled for release | Prison Pengkalan Chepa, 33.7 y ± 6.7; Prison Kajang, 37.1 y ± 7.0 | 100% male | Prison Pengkalan Chepa: Malay, 95.2%; Indian, 4.8%; Prison Kajang: Malay, 73.3%; Indian, 20.0%; Chinese, 6.7% | DSM-IV for opioid dependence |
| Westerberg et al., 2016 [67] | NM, US | 960 | Cohort (2011–2013) | Jail | Incarcerated persons released between July and December 2011 | NA | Male, 73.8%; female, 26.2% | African-American, 6.0%; Hispanic, 49.7%; Native American, 15.0%; white, 25.6%; unknown/other, 3.7% | MMT previous incarceration |
| Marsden et al., 2017 [68] | England | 12,260 | Cohort (2010–2013) administrative data | Prison | Incarcerated persons scheduled to be released | OAT in prison, 34.6 y ± 7.1; no OAT in prison, 34.6 y ± 8.0 | OAT in prison: Male, 75.9%; female, 24.1%; no OAT in prison: male, 80.7%; female, 19.3% | NA | Clinical diagnosis of OUD |
| Huang et al., 2011 [69] | Taiwan | 4,357 | Cohort (2007–2008) administrative data | Prison | Incarcerated persons released on 16 July 2007 | Male, 38 y (20–74); female, 31 y (21–58) | Male, 88%; female, 12% | NA | Self-reported history of heroin injection |

*(Continued)*

**Table 1.** (Continued)

| Source | Study location | N | Design (period) | Setting | Characteristics of study population | | | | |
|---|---|---|---|---|---|---|---|---|---|
| | | | | | Population | Age (years)* | Sex | Ethnicity (%) | Addiction criteria |
| Lee et al., 2012 [70] | NY, US | 140 | Cohort (2006–2008) | Jail | Post-release patients from NYC Department of Correction | Jail referrals, 41 y (21–52); community referrals, 42 y (25–67) | Jail referrals: male, 97%; female, 3%; community referrals: male, 78%; female, 22% | Jail referrals: African-American, 19%; Hispanic, 66%; non-Hispanic white, 15%; community referrals: African-American, 13%; Hispanic, 34%; non-Hispanic white, 53% | DSM-IV for opioid dependence |
| Macswain et al., 2014 [71] | Canada | 856 | Cohort (2006–2008) | Prison | Incarcerated persons with problematic opioid use | MMT in prison but discontinued post-release, 34.3 y ± 8.1; MMT in prison and continued post-release, 35.3 y ± 8.6; non-MMT treated, 34.6 y ± 8.3 | 100% male | MMT discontinued post-release and Aboriginal ancestry, 15.0%; MMT continued post-release and Aboriginal ancestry, 16.8%; non-MMT treated and Aboriginal ancestry, 15.4% | MMT prior to incarceration |
| Fox et al., 2014 [72] | NY, US | 135 | Cohort (2009–2013) | Prison | Incarcerated persons recently released from prison (≤90 days before initial visit) | 42.1 y ± 10.5 | Male, 97.0%; female, 3.0% | Hispanic, 50.4%; non-Hispanic black, 42.2%; non-Hispanic other, 7.4% | DSM-IV for opioid dependence |
| Riggins et al., 2017 [73] | NY, US | 306 | Cohort (NA) | Prison and jail | HIV-positive patients who were recently incarcerated (last 30 days) | 44.6 y ± 8.5 | Male, 67.2%; female, 32.8% | Hispanic, 22.0%; non-Hispanic black, 51.2%; non-Hispanic other, 3.3%; non-Hispanic white, 22.3% | Self-reported opioid use (past 30 days) |
| Bird et al., 2016 [74] | Scotland | 2,273 | Cohort (2006–2013) administrative data | Prison | Incarcerated persons with data related to ORD | Baseline, <35 y (52.8%); during NNP, <35 y (38.6%) | Baseline: Male, 80.9%; female, 19.1%; during NNP: male, 76.1%; female, 23.9% | NA | Self-reported risk of opioid overdose at release |
| Bird et al., 2017 [75] | Scotland | 4,124 | Cohort (2006–2015) administrative data | Prison | Incarcerated persons with data related to ORD | NA | NA | NA | Self-reported risk of opioid overdose at release |
| Springer et al., 2010 [76] | CT, US | 23 | RCT (2004–2010) | Prison | HIV-positive incarcerated persons (up to 90 days pre-release) | 46.4 y (mean) | Male, 78%; female, 22% | Black, 39%; Hispanic, 52%; white, 9% | DSM-IV for opioid dependence |
| Springer et al., 2012 [77] | CT, US | 94 | RCT (2005–2010) | Prison | HIV-positive incarcerated persons transitioning to the community | BPN/NLX, 45.6 y ± 6.0; no BPN/NLX, 46.5 y ± 7.5 | BPN/NLX: male, 88%; female, 12%; no BPN/NLX: male, 75%; female, 25% | BPN/NLX: white, 12.0%; black, 32.0%; Hispanic, 56.0%; no BPN/NLX: white, 25.0%; black, 54.6%; Hispanic, 20.4% | DSM-IV for opioid dependence |
| Lobmaier et al., 2010 [78] | Norway | 46 | RCT (2005–2007) | Prison | Incarcerated persons with at least 2 months of sentence time remaining | 35.1 y ± 7.0 | Male, 93.2%; female, 6.8% | NA | DSM-IV for opioid dependence |

*(Continued)*

**Table 1.** (Continued)

| Source | Study location | N | Design (period) | Setting | Characteristics of study population | | | | |
|---|---|---|---|---|---|---|---|---|---|
| | | | | | Population | Age (years)* | Sex | Ethnicity (%) | Addiction criteria |
| Parmar et al., 2017 [79] | England | 1,557 | RCT (2012–2014) | Prison | Incarcerated persons (up to 3 months pre-release) | 18–24 y (5%); 25–34 y (50%); 35–44 y (39%); ≥45 y (6%) | Male, 98%; female, 2% | NA | Self-reported history of heroin use by injection |

*Values given as category (percentage), mean ± SD, median (IQR), or range.

BPN, buprenorphine; DSM-IV, Diagnostic and Statistical Manual of Mental Disorders, 4th edition; MMT, methadone maintenance treatment; NA, not available; NLX, naloxone; NNP, National Naloxone Program; NYC, New York City; ORD, opioid-related death; OAT, opioid agonist treatment; OST, opioid substitution therapy; OUD, opioid use disorder; RCT, randomized control trial; RIDOC, Rhode Island Department of Corrections; TAU, treatment as usual; XR-NTX, injectable extended-release naltrexone.

primary data, including 22 randomized control trials (RCTs) (median of 198 [range 15–1,557] participants), 13 longitudinal/observational studies (median of 140 [range 27–960] participants), and 3 non-randomized clinical trials (median of 44 [range 27–382] participants) (Table 1).

Studies were grouped according to when the opioid use intervention was provided relative to the incarceration period (pre-incarceration, during incarceration, or post-incarceration). Twenty-one studies evaluated the impact of opioid use interventions provided during a continuum of treatment (before, during, and after incarceration), 14 studies reported the effect of opioid use interventions offered to individuals while incarcerated, and 11 studies evaluated the effects of opioid use interventions post-incarceration. The last group included 3 studies that addressed the impact of NLX, a preventive intervention to reduce opioid-related overdose harms and mortality (Table 1). Interventions included various forms of pharmacological treatment for OUD (including methadone, BPN, BPN/NLX, naltrexone) and the distribution of NLX kits to at-risk individuals before their released into the community, to decrease the occurrences of opioid-related overdose and mortality.

Thirty-two studies were conducted in prisons, 6 in jails, and 8 in both types of correctional facilities. Incarcerated persons sentenced to terms longer than 1 year are generally sent to prison, while jails are characterized by a rapid turnover of both sentenced and non-sentenced arrested persons. This specificity should be taken into account when evaluating the results of OAT delivered in long-term versus short-term incarceration facilities.

## Opioid use interventions during a continuum of treatment (i.e., before, during, and after incarceration)

**Study characteristics.** Twenty-one studies reported the impact of an opioid intervention provided during a continuum of treatment prior to, during, and after incarceration (Table 2). Fourteen of these studies were experimental [37–50], and 7 were observational [35,36,51–55]. Five studies evaluated the impact of receiving opioid substitution therapy—primarily methadone—while incarcerated compared to treatment post-incarceration [35,36,51,53,54]. Seven studies evaluated the impact of receiving methadone maintenance treatment (MMT) while incarcerated, with participants continuing or discontinuing MMT post-release [38,39,43–46,52]. Four studies evaluated the impact of receiving BPN maintenance treatment (BMT) while incarcerated, with participants continuing or discontinuing BMT post-release [37,40,42,47], and 5 studies evaluated the impact of receiving injectable extended-release naltrexone (XR-NTX) while incarcerated compared to post-release [41,48–50,55].

**Table 2. Description of interventions and outcomes of opioid use intervention studies during a continuum of treatment (before, during, and after incarceration), 2008–2019.**

| Source | Intervention | Sample size | Outcomes/conclusions |
|---|---|---|---|
| **Non-specified opioid agonist treatment** | | | |
| Degenhardt et al., 2014 [35] | OAT in prison vs. post-release | N = 16,453 | 76.5% of individuals received OAT while incarcerated |
| | | | **Crude mortality rates at 4 weeks post-release** |
| | | | Among those retained in OAT post-release: 8.8 per 1,000 PY (95% CI 5.0–14.3) |
| | | | Among those not in OAT post-release: 36.7 per 1,000 PY (95% CI 28.8–45.9) |
| | | | OAT exposure by 4 weeks post-release reduced hazard of death by 75% (AHR = 0.25, 95% CI 0.12–0.53) |
| | | | OAT receipt in prison had a short-term protective effect that decayed quickly across time |
| | | | Lowest post-release mortality observed among those continuously retained in OAT post-release |
| Larney et al., 2014 [36] | OAT in prison | N = 16,715 | 76.9% received OAT while incarcerated |
| | | | Mortality of opioid-dependent incarcerated persons was significantly lower among those receiving OAT in prison |
| | | | Hazard of all-cause death was 74% lower among those receiving OAT in prison vs. those opioid-dependent not in OAT (AHR = 0.26, 95% CI 0.13–0.50) |
| | | | Hazard of unnatural death was 87% lower among those receiving OAT in prison vs. those opioid-dependent not in OAT (AHR = 0.13, 95% CI 0.05–0.35) |
| | | | Hazard of all-cause death during first 4 weeks of incarceration was 94% lower among those receiving OAT in prison vs. those opioid-dependent not in OAT (AHR = 0.06, 95% CI 0.01–0.48) |
| | | | Hazard of unnatural death during first 4 weeks of incarceration was 93% lower among those receiving OAT in prison vs. those opioid-dependent not in OAT (AHR = 0.07, 95% CI 0.01–0.53) |
| Larney et al., 2012 [51] | OAT in prison vs. post-release | N = 375 (OAT = 331) | 80% of participants started OAT in prison, with median treatment duration of 5.5 months |
| | | | Median duration of post-release OAT was 63 days |
| | | | Participants retained in OAT post-release had lower risk of incarceration (HR = 0.80, 95% CI 0.71–0.90, p < 0.001) |
| Larney et al., 2016 [53] | OAT in prison | N = 8,577 | 82% retention in OAT treatment until release |
| | | | 90% of participants received OAT prescription prior to release |
| | | | 94% of participants with a prescription presented to a community clinic within 48 hours of release |
| Bird et al., 2015 [54] | Before vs. after OAT in prison | N = 131,472 | **Before prison-based OAT (1996–2002)** |
| | | | 305 DRDs within 12 weeks post-release, with 175 deaths (57%) within 14 days post-release |
| | | | 3.8 deaths per 1,000 releases (95% CI 3.4–4.2) |
| | | | **After prison-based OAT (2003–2007)** |
| | | | 154 DRDs within 12 weeks post-release, with 56% of deaths within 14 days post-release |
| | | | 2.2 deaths per 1,000 releases (95% CI 1.8–2.5) |
| | | | When DRD in each period was compared, a significant decrease was identified: 1.6 DRD per 1,000 releases (95% CI 1.0–2.2; p < 0.001) |
| **Methadone maintenance treatment** | | | |
| Gordon et al., 2012 [38] | RCT (3 groups): Co vs. C+T vs. C+M | N = 211 (Co, 70; C+T, 70; C+M, 71) | **Impact of C+M** |
| | | | More likely to enter prison treatment for opioid use vs. Co (OR = 10.6, 95% CI 2.6–42.8; p < 0.001) |
| | | | More likely to complete prison treatment for opioid use vs. Co (OR = 3.5, 95% CI 1.5–8.3; p < 0.01) |
| | | | **Impact of C+T** |
| | | | More likely to complete prison treatment for opioid use vs. Co (OR = 3.6, 95% CI 1.5–8.4; p < 0.01) |

*(Continued)*

**Table 2.** (Continued)

| Source | Intervention | Sample size | Outcomes/conclusions |
|---|---|---|---|
| Gordon et al., 2008 [39] | RCT (3 groups): Co vs. C+T vs. C+M | N = 201 (Co, 63; C+T, 68; C+M, 70) | **Impact of C+M (6 months post-release)** |
| | | | More likely to start addiction treatment in prison vs. Co (*p* = 0.001) and C+T (*p* = 0.046) |
| | | | Remained more days in addiction treatment vs. Co and C+T (*p* < 0.001; SE: 0.31) |
| | | | Reported fewer days of heroin use vs. Co (*p* = 0.009; SE: 0.24) |
| | | | Reported fewer days of criminal activity vs. Co (*p* = 0.025; SE: 0.29) |
| | | | Co more likely to test positive for opioid vs. C+M (OR = 4.68, 95% CI 1.77–12.43, *p* = 0.002) |
| Kinlock et al., 2008 [43] | RCT (3 groups): Co vs. C+T vs. C+M | N = 197 (Co, 63; C+T, 66; C+M, 68) | **Impact of C+M (90 days post-release)** |
| | | | More likely to enter community treatment for opioid use vs. Co (OR = 61.7, 95% CI 16.0–237.7, *p* < 0.001) |
| | | | Less likely to be re-incarcerated vs. Co (OR = 0.36, 95% CI 0.14–0.91, *p* < 0.05) |
| | | | Less likely to report heroin use vs. Co (OR = 0.36, 95% CI 0.16–0.81, *p* < 0.05) |
| | | | Less likely to engage in criminal activity vs. Co (OR = 0.34, 95% CI 016–0.73, *p* < 0.01) |
| Kinlock et al., 2009 [44] | RCT (3 groups): Co vs. C+T vs. C+M | N = 204 (Co, 64; C+T, 69; C+M, 71) | **Impact of C+M (12 months post-release)** |
| | | | Remained more days in addiction treatment vs. Co and C+T (*p* < 0.001) |
| | | | Co more likely to have a positive opioid test vs. C+M (OR = 7.1, 95% CI 1.4–11.3, *p* < 0.001) |
| | | | Less likely to have a positive cocaine test vs. Co (*p* < 0.001) and C+T (*p* < 0.05) |
| Kinlock et al., 2013 [45] | RCT (sub-sample) | N = 67 | **Impact of C+M** |
| | | | 74.6% of participants completed in-prison MMT treatment |
| | | | Employment 3 years prior to incarceration predicted completing 1 year of community treatment post-release (*p* = 0.001) |
| | | | Participants who completed 1 year of community-based MMT reported working over twice as many days as other participants (*p* = 0.003) |
| McKenzie et al., 2012 [46] | RCT: MMT prior to release vs. MMT referral post-release | N = 62 (arm 1, 21; arm 2, 32; arm 3, 9) | Arm 1: MMT pre- and post-release + payment of treatment costs (12 weeks) |
| | | | Arm 2: referral to MMT upon release + payment of treatment costs (12 weeks) |
| | | | Arm 3: referral to MMT upon release (no financial assistance) |
| | | | Arm 1 more likely to enter MMT within 30 days post-release (86%, 41%, and 22% for arm 1, 2, and 3, respectively, *p* < 0.001) |
| | | | Arm 1 entered MMT post-release in fewer days (2, 9, and 5 days for arm 1, 2, and 3, respectively, *p* = 0.03) |
| | | | In the last 30 days, arm 1 reported less heroin use (3, 18, and 4 days for arm 1, 2, and 3, respectively, *p* = 0.008), less use of other opiates (0, 6, and 1 day, *p* = 0.09), less crack/cocaine use (4, 13, and 6 days, *p* = 0.05), and less injection drug use (2, 12, 3 days, *p* = 0.06) |
| Farrell-MacDonald et al., 2014 [52] | MMT post-release: continued MMT (MMT-C) vs. discontinued MMT (MMT-T) vs. no MMT (MMT-N) | N = 137 | MMT-C group had lower risk of return to custody vs. MMT-N (HR = 0.35, 95% CI 0.13–0.90, *p* < 0.05) |
| **Buprenorphine maintenance treatment** | | | |
| Gordon et al., 2014 [37] | RCT (4 groups): B+OTP, B+CHC, C+OTP, C+CHC | N = 211 (B+OTP, 52; B+CHC, 52; C+OTP, 54; C+CHC, 53) | BPN group (B+OTP and B+CHC) more likely to enter prison treatment vs. counseling only (C+OTP and C+CHC) (AOR = 2.8, 95% CI 0.3–5.7, *p* = 0.006) |
| | | | BPN group more likely to enter community treatment vs. counseling only (AOR = 1.5, 95% CI 1.1–2.1, *p* = 0.01) |
| Gordon et al., 2017 [40] | RCT (4 groups): B+OTP, B+CHC, C+OTP, C+CHC | N = 211 | Higher post-release addiction treatment retention rates among participants who initiated BPN in prison (65.9 days, SE = 12.2) vs. initiation post-release (21.8 days, SE = 7.6, *p* = 0.005) |

(*Continued*)

**Table 2.** (Continued)

| Source | Intervention | Sample size | Outcomes/conclusions |
|---|---|---|---|
| Gordon et al., 2018 [42] | RCT (4 groups): B+OTP, B+CHC, C+OTP, C+CHC | N = 199 | No statistically significant differences in BPN treatment initiation pre- vs. post-incarceration on the following variables: proportion of individuals arrested, mean number of arrests, time to first arrest |
| Zaller et al., 2013 [47] | Clinical trial: BPN/NLX (post-release vs. in jail) | N = 44 (post-release, 32; in jail, 12) | Time until post-release appointment for addiction treatment was lower among in-jail (3.9 days) vs. post-release group (8.8 days; $p = 0.1$) |
| | | | Post-release treatment duration was higher for in-jail vs. post-release group: 24 vs. 9 weeks ($p = 0.007$) |
| | | | After 6 months, retention was higher for in-jail (83%) vs. post-release group (34%; $p = 0.005$) |
| | | | Initiating BPN/NLX prior to release from incarceration increased engagement and retention in community-based treatment |
| | | | Past 30 days heroin use was higher among those receiving BPN/NLX post-release (34%) vs. in jail (0%; $p = 0.08$) |
| | | | Past 30 days alcohol use was higher among those receiving BPN/NLX post-release (47%) vs. in jail (10%; $p = 0.14$) |
| | | | Past 30 days injection drug use was higher among those receiving BPN/NLX post-release (39%) vs. in jail (0%; $p = 0.05$) |
| | | | Past 30 days arrest was higher among those receiving BPN/NLX post-release (24%) vs. in jail (0%; $p = 0.08$) |
| **Injectable extended-release naltrexone** | | | |
| Gordon et al., 2015 [41] | Clinical trial: XR-NTX | N = 27 | 37% of participants completed all 6 monthly post-release XR-NTX injections |
| | | | Among participants who completed all 6 injections ($n = 10$), none reported opioid use, re-arrest, or re-incarceration during the study. |
| | | | Among participants who did not complete all 6 injections ($n = 16$), 62.5% reported opioid use during the study (10/16), 31.3% reported re-arrest (5/16), and 18.8% reported re-incarceration (3/16) |
| | | | Results were not statistically significant. |
| Lee et al., 2016 [48] | RCT (2 groups): XR-NTX vs. OAT (MMT/BPN) 1 week prior to release | N = 308 | **During the 24-week treatment phase, participants assigned to XR-NTX (compared to those assigned to usual treatment)** |
| | | | Longer median time to opioid relapse (10.5 vs. 5.0 weeks, $p < 0.001$; HR = 0.49, 95% CI 0.36–0.68) |
| | | | Lower rate of opioid relapse (43% vs. 64% of participants, $p < 0.001$; OR = 0.43, 95% CI 0.28–0.65) |
| | | | Higher rate of opioid-negative urine samples (74% vs. 56%, $p < 0.001$; OR = 2.30, 95% CI 1.48–3.54) |
| Friedmann et al., 2018 [49] | RCT (2 groups): pre-release vs. post-release XR-NTX | N = 15 | **Pre-release XR-NTX group had better treatment retention rate vs. post-release XR-NTX group** |
| | | | 100% of participants received the first XR-NTX injection vs. 67% of post-release group |
| | | | 78% of participants received more than 1 injection vs. 17% of post-release group |
| | | | 22% of participants received all 6 injections vs. no participants in the post-release group |
| | | | **Pre-release XR-NTX group had greater abstinence vs. post-release XR-NTX group** |
| | | | Confirmed abstinence 4 weeks post-release (OR = 5.6, 95% CI 0.8–37.9, $p = 0.08$) |
| | | | **Pre-release XR-NTX group had increased time to relapse vs. post-release XR-NTX group** |
| | | | 9 weeks vs. 5 weeks |
| Soares et al., 2018 [50] | RCT (2 groups): XR-NTX vs. OAT (MMT/BPN) 1 week prior to release | N = 297 | No significant difference in overall healthcare utilization (IRR = 0.88, 95% CI 0.63–1.23, $p = 0.45$) |
| | | | XR-NTX group had fewer medical/surgical hospital admissions during the treatment phase (IRR = 0.37, 95% CI 0.16–0.88, $p = 0.02$) and throughout the course of the study (IRR = 0.55, 95% CI 0.30–1.00, $p = 0.05$) |

*(Continued)*

**Table 2.** (Continued)

| Source | Intervention | Sample size | Outcomes/conclusions |
|---|---|---|---|
| Lincoln et al., 2018 [55] | Cohort: pre-release vs. post-release XR-NTX | N = 67 | **Receiving XR-NTX prior to release from jail increased treatment retention vs. post-release XR-NTX** |
| | | | 4 weeks post-release: 55% vs. 25% |
| | | | 8 weeks post-release: 36% vs. 25% |
| | | | 24 weeks post-release: 21% vs. 15% |

AHR, adjusted hazard ratio; AOR, adjusted odds ratio; B+OTP, BPN in prison and continued at an opioid treatment program; B+CHC, BPN in prison and continued at a community health center; BPN, buprenorphine; C+M, counseling and MMT in prison + referral to MMT upon release; C+OTP, counseling in prison and initiation of BPN at an opioid treatment program; C+CHC, counseling in prison and initiation of BPN a community health center; C+T, counseling in prison + referral to MMT upon release; Co, counseling in prison; DRD, drug-related death; HR, hazard ratio; IRR, incidence rate ratio; MMT, methadone maintenance treatment; NLX, naloxone; OR, odds ratio; OAT, opioid agonist treatment; PY, person-years; RCT, randomized control trial; XR-NTX, injectable extended-release naltrexone.

**Non-specified opioid substitution treatment.** Several Australian studies evaluated the impact of OAT provision while incarcerated on post-release OAT retention and related outcomes [35,36,51,53]. A 2012 study conducted by Larney et al. [51] found that among 375 incarcerated, heroin-using men from New South Wales, those who received OAT while in prison and continued OAT post-release reduced their risk of re-incarceration by 20% (hazard ratio [HR] = 0.80, 95% CI 0.71–0.90, p = 0002), compared to those who discontinued OAT post-release. Larney et al. published another study in 2016 [53] to evaluate administrative data from 8,577 incarcerated persons who were in prison between 2007 to 2013 and experienced changes in the clinical governance of the New South Wales OAT program in 2011. The authors identified statistically significant higher rates of OAT retention while incarcerated (82%), higher rates of OAT prescription prior to release (90%), and higher rates of presentation to a community OAT clinic within 48 hours post-release (94%) among incarcerated persons following changes to the OAT program.

Larney et al. and Degenhardt et al. both examined the relationship between receipt of OAT and mortality risk, based on administrative data from a cohort of over 16,000 incarcerated persons in New South Wales, Australia [35,36]. In Larney et al.'s study [36], the majority of individuals received OAT while incarcerated (76.9%), and this intervention reduced the hazard of all-cause and unnatural death by 74% (adjusted hazard ratio [AHR] = 0.26, 95% CI 0.13–0.50) and 87% (AHR = 0.13, 95% CI 0.05–0.35), respectively, compared to those out of OAT. Moreover, the hazard of all-cause death and unnatural death during the first 4 weeks of incarceration was reduced by 94% (AHR = 0.06, 95% CI 0.01–0.48) and 93% (AHR = 0.07, 95% CI 0.01–0.53), respectively, during periods on OAT compared to those out of OAT. Degenhardt et al. [35] also found that the majority of individuals received OAT while incarcerated (76.5%), determining that OAT exposure by 4 weeks post-release reduced the hazard of death by 75% (AHR = 0.25, 95% CI 0.12–0.53). Post-release mortality rates (MRs) were lowest among those continuously engaged in OAT at 4 weeks post-release (8.8 per 1,000 person-years [PY], 95% CI 5.0–14.3) and highest among those not receiving OAT 4 weeks post-release (36.7 per 1,000 PY, 95% CI 28.8–45.9).

Bird et al. [54] evaluated a large administrative dataset (n = 131,427) of all incarcerated persons in Scottish prisons between 1996 and 2007 to determine the effects of receiving OAT on drug-related deaths post-release. The drug-related death rate within 12 weeks after prison release fell from 3.8 per 1,000 releases (95% CI 3.4–4.2) to 2.2 per 1,000 releases (95% CI 1.8–2.5) following the implementation of a universal prison-based OAT policy for incarcerated

individuals with OUD between 2003 and 2007. Furthermore, the decrease in MR of 1.6 per 1,000 (95% CI 1.0–2.2) was found to be highly statistically significant ($p < 0.001$).

**Methadone maintenance treatment.** Gordon et al. and Kinlock et al. [38,39,43–45] conducted an RCT comparing 3 groups of heroin-dependent incarcerated persons in a Baltimore prison ($n = 211$), with one group receiving only counseling while incarcerated, a second group receiving counseling and referral to MMT upon release, and a third group receiving both counseling and MMT while incarcerated and MMT referral post-release. Short-term results at 3 months post-release [43] and long-term results at 6 and 12 months post-release [39,44] found that participants who received both MMT and counseling while in prison displayed higher adherence and retention to opioid-related community-based treatment, lower rates of illicit opioid use, and lower re-incarceration rates, compared to those who received counseling only during incarceration, independent of whether they were referred or not to MMT upon prison release. Specifically, the results showed that participants who initiated MMT while in prison were more likely to engage in community-based treatment within 90 days post-release ($p < 0.01$), remained in addiction treatment for a greater number of days at 6 and 12 months post-release ($p < 0.001$), and were less likely to test positive for opioid use at 3, 6, and 12 months post-release ($p = 0.014$, $p = 0.009$, and $p = 0.001$, respectively) compared to the counseling only group [39,43,44]. Participants who received both MMT and counseling while in prison were also less likely to report engagement in criminal activities at 3 months ($p = 0.005$) and 6 months ($p = 0.025$) post-release, compared to the other groups [39,43]. Further secondary analyses conducted by Kinlock et al. [45] found that participants who completed 1 year of community-based MMT post-release worked twice as many days (122 days versus 56 days) relative to those who did not complete 1 year of MMT after prison release.

McKenzie et al. [46] reported findings from an RCT ($n = 62$) assessing 3 conditions: one group of incarcerated persons who received MMT while in prison and a 12-week financial subsidy to continue treatment upon release (arm 1), a second group referred to MMT after prison release and provided with a 12-week financial subsidy (arm 2), and a third group referred to MMT with no provision of a financial subsidy upon release. Participants who received MMT while in prison were more likely to initiate MMT within 30 days post-prison release (86%, 41%, and 22% for arm 1, 2, and 3, respectively, $p < 0.001$). Among those who entered MMT after prison release, those who received MMT in prison entered within fewer days (2, 9, and 5 days for arm 1, 2, and 3, respectively, $p = 0.03$). Those who received MMT pre-release reported, for the last 30 days, less heroin use (3, 18, and 4 days for arm 1, 2, and 3, respectively, $p = 0.008$), less use of other opiates (0, 6, and 1 day, $p = 0.09$), less crack/cocaine use (4, 13, and 6 days, $p = 0.05$), and less injection drug use (2, 12, and 3 days, $p = 0.06$). Farrell-MacDonald et al. [52] examined administrative data from a cohort of 137 incarcerated women with OUD at a Canadian federal facility, comparing 3 groups: group 1 initiated MMT while incarcerated and continued MMT post-release, group 2 initiated MMT while incarcerated but terminated treatment post-release, and group 3 were women with OUD who did not participate in MMT while incarcerated. Women with OUD who received MMT while incarcerated and continued MMT post-release had a 65% lower risk of returning to custody during the 6-year follow-up period (HR = 0.35, 95% CI 0.13–0.90) compared to the women who did not participate in MMT while incarcerated.

**Buprenorphine maintenance treatment.** Gordon et al. [37,40,42] conducted an RCT ($n = 211$) comparing incarcerated persons with histories of heroin dependence. Participants were randomly assigned into 1 of 4 conditions: one group received in-prison BMT and continued treatment at an OAT program post-release, a second group received in-prison BMT and continued treatment at a community health center post-release, a third group received counseling only during their incarceration and treatment at an OAT program post-

release, and a fourth group received counseling only during their incarceration and treatment at a community health center post-release. The authors determined that those who received BMT while incarcerated and continued at an OAT program post-release were more likely than the counseling only groups to enter prison treatment (adjusted odds ratio [AOR] = 2.8, 95% CI 1.3–5.7, $p = 0.006$) and to enter community treatment post-release (AOR = 1.5, 95% CI 1.1–2.1, $p = 0.01$) [37]. Participants who received BPN while incarcerated had also higher rates of treatment retention at 12 months post-release compared to those who initiated BPN after release ($p = 0.005$) [40]. However, the study did not discern any statistically significant effects of the receipt of BPN treatment during incarceration on days of heroin use, positive opioid and cocaine urine screening test results, and re-arrest rate at 12 months of follow-up [40,42].

Zaller et al. [47] conducted a clinical trial ($n = 44$) to evaluate the provision of BPN/NLX to incarcerated persons from the Rhode Island Department of Corrections (RIDOC) diagnosed with opioid dependence, specifically comparing those who initiated OAT during incarceration and post-release. Participants who received BPN/NLX while incarcerated started treatment earlier post-release (3.9 versus 8.8 days, $p = 0.1$) and had longer retention in OAT after release (24 versus 9 weeks, $p = 0.007$). The authors also determined that treatment retention was higher at 6 months post-release among those who started BPN/NLX treatment in prison relative to those who initiated BPN/NLX post-release ($p = 0.005$). Furthermore, rates of heroin use and re-incarceration were higher among those who started BPN treatment post-release, compared to participants who initiated BPN/NLX during their incarceration period.

**Injectable extended-release naltrexone.**   Six studies evaluated the impact of XR-NTX on relapse into illicit opioid use, treatment retention, re-incarceration, and healthcare utilization [41,48–50,55]. Gordon et al. [41] conducted a clinical trial ($n = 27$) to evaluate the relationship between XR-NTX adherence and re-incarceration among persons with OUD prior to incarceration. The authors found that participants who completed XR-NTX treatment (involving 1 injection prior to release and 6 monthly injections post-release) were less likely to use illicit opioids during the course of the study, compared to those who did not complete the treatment (i.e., received <6 injections, $p = 0.003$). In addition, participants who did not complete the treatment were more likely to be re-arrested (31.3% versus 0%) or re-incarcerated (18.8% versus 0%) relative to those who did complete the treatment; however, the differences in recidivism outcomes—including re-arrest and re-incarceration rates—were not statistically significant.

Lee et al. [48] conducted a 5-site RCT ($n = 308$) comparing previously incarcerated participants receiving XR-NTX ($n = 153$) with those receiving treatment as usual, including BPN or MMT ($n = 155$). After the 24-week treatment and follow-up period, the participants assigned to receive XR-NTX had a longer median time to opioid relapse (10.5 versus 5.0 weeks, $p < 0.001$; HR = 0.49, 95% CI 0.36–0.68), a lower rate of opioid relapse (43% versus 64%, $p < 0.001$; OR = 0.43, 95% CI 0.28–0.65), and a higher rate of opioid-negative urine samples (74% versus 56%, $p < 0.001$; OR = 2.30, 95% CI 1.48–3.54). However, at week 78 (approximately 1 year after the end of the treatment phase), the rates of opioid-negative urine samples were equal (46% in each group, $p = 0.91$). After 78 weeks of follow-up, there were no overdose events (fatal or nonfatal) among participants allocated to XR-NTX, and 7 among participants assigned to treatment as usual ($p = 0.02$).

Friedmann et al. [49] conducted an RCT study with 15 incarcerated persons from RIDOC comparing one group of participants who received 1 XR-NTX injection prior to release followed by 5 monthly treatments post-release and a second group where participants received 6 XR-NTX injections in the community exclusively. The authors found that participants, who received their first dose of XR-NTX treatment prior to release, exhibited greater abstinence rates as determined by a higher proportion of self-reported opioid-free days at 2

weeks and 1 month post-release, relative to participants who received 6 doses of XR-NTX post-release (OR = 5.6, 95% CI 0.8–37.9, $p$ = 0.08).

Lincoln et al. [55] evaluated data from 67 incarcerated persons released from a Massachusetts county jail, comparing those who initiated XR-NTX treatment prior to release ($n$ = 47) with those who initiated XR-NTX treatment post-release ($n$ = 20). The authors found that receipt of XR-NTX injection prior to release was associated with increased XR-NTX treatment retention rates at 4, 8, and 24 weeks post-release (55% versus 25%, 36% versus 25%, and 21% versus 15%, respectively), compared to incarcerated persons who commenced the treatment post-release.

Soares et al. [50] conducted additional analysis of the study described by Lee and colleagues [48], evaluating the impact of XR-NTX treatment, compared to treatment as usual, on healthcare utilization among adults with a history of OUD involved in the criminal justice system. Although the authors did not identify any statistically significant differences in re-incarceration between those who received XR-NTX compared to those who received standard OAT, individuals who received XR-NTX experienced fewer medical and/or surgical hospital admissions during the 6-month treatment phase (incidence rate ratio [IRR] = 0.37, 95% CI 0.16–0.88, $p$ = 0.02) and at 12 months post-treatment (IRR = 0.55, 95% CI 0.30–1.00, $p$ = 0.05) relative to the treatment as usual group.

### Opioid use interventions during incarceration

**Study characteristics.** Fourteen studies assessed the impact of OAT provision while incarcerated (Table 3). Seven of these studies were experimental [56–62], and 7 were observational [26,63–68]. Eight studies evaluated the impact of receiving OAT—primarily methadone—while incarcerated compared to opioid detoxification [26,60–64,67,68], and 2 studies evaluated the impact of methadone dose at the time of release [65,66]. Three RCTs compared the efficacy of BPN and methadone [57–59], and 1 RCT compared BPN and dihydrocodeine for opiate detoxification within a UK prison setting [56].

**Non-specified opioid substitution treatment.** Two studies assessed the impact of receiving OAT while incarcerated on opioid-related overdose mortality outcomes post-release, with both identifying large and significant reductions among incarcerated individuals with OUD who received OAT in prison/jail, compared to those who did not receive OAT during their incarceration period [26,68]. Green et al. [26] assessed the impact of a new model of screening and treatment for OUD started at RIDOC in 2016, involving the initiation and continuation of OAT in incarcerated individuals. Between 2016 and 2017, the authors identified a 60.5% reduction in opioid-related overdose deaths post-incarceration (26 deaths versus 9 deaths; risk ratio = 0.4, 95% CI 0.18–0.81, $p$ = 0.01).

Marsden et al. [68] evaluated data from a large national cohort of 12,260 incarcerated persons diagnosed with OUD from 39 adult prisons in England, 6,662 of whom received OAT while incarcerated. The study found substantial reductions in all-cause mortality (AHR = 0.25, 95% CI 0.09–0.64) and drug-related mortality (85%; AHR = 0.15, 95% CI 0.04–0.52) within the first month of release associated with OAT exposure at the time of release. Receipt of prison-based OAT during incarceration was also associated with an increased likelihood of entering community drug treatment within 4 weeks post-release (OR = 2.47, 95% CI 2.31–2.65).

**Methadone maintenance treatment.** Multiple studies showed that individuals with OUD who engage with MMT while incarcerated exhibit better opioid-use-related health outcomes, both during their incarceration period and post-release. Brinkley-Rubinstein et al. [61] reported 12-month findings from an RCT involving 179 incarcerated persons from Rhode

**Table 3. Description of intervention and related outcomes of opioid use interventions during incarceration, 2008–2019.**

| Source | Intervention | Sample size | Outcomes/conclusions |
|---|---|---|---|
| **Non-specified opioid agonist treatment** | | | |
| Green et al., 2018 [26] | Cohort: OAT in prison | N = 35 | Overdose deaths in 2016: 179, with 26 recently incarcerated (14.5%) |
| | | | Overdose deaths in 2017: 157, with 9 recently incarcerated (5.7%) |
| | | | Reduction in overdose-related mortality after the release of new model of screening and OAT treatment in 2016: 60.5% (risk ratio = 0.4, 95% CI 18.4–80.9, $p = 0.01$) |
| Marsden et al., 2017 [68] | Cohort: OAT vs. no OAT in prison | N = 12,260 | All-cause mortality lower among OAT-exposed vs. unexposed group 4 weeks post-release: 0.93 per 100 PY vs. 3.67 per 100 PY (AHR = 0.25, 95% CI 0.09–0.64) |
| | | | Drug-related poisoning deaths lower among OAT-exposed vs. unexposed group 4 weeks post-release: 0.47 per 100 PY vs. 3.06 per 100 PY (AHR = 0.15, 95% CI 0.04–0.52) |
| | | | No group difference in mortality risk following the first month |
| | | | OAT-exposed group more likely to enter addiction treatment during first month post-release (OR = 2.47, 95% CI 2.31–2.65) |
| | | | Prison-based OAT associated with 75% reduction in all-cause mortality and 85% reduction in fatal drug-related poisoning during first month post-release |
| **Methadone maintenance treatment** | | | |
| Rich et al., 2015 [60] | RCT (2 groups): MMT vs. forced tapered withdrawal during incarceration | N = 283 | MMT access (within 1 month post-release) higher among in-prison MMT vs. forced withdrawal group (HR = 2.04, 95% CI 1.48–2.80, $p < 0.001$) |
| | | | MMT initiation (within 1 month post-release) higher among in-prison MMT vs. forced withdrawal group (HR = 6.61, 95% CI 4.00–10.91, $p < 0.001$) |
| Brinkley-Rubinstein et al., 2018 [61] | RCT (2 groups): MMT vs. forced tapered withdrawal during incarceration | N = 179 | **Impact at 12 months post-release of tapered withdrawal from methadone vs. MMT in prison** |
| | | | Heroin use during prior 30 days: 39.2% vs. 24.2% (OR = 2.02, 95% CI 1.01–4.04, $p < 0.05$) |
| | | | Injection drug use during prior 30 days: 39.2% vs. 18.0% (OR = 2.95, 95% CI 1.43–6.06, $p < 0.05$) |
| | | | Nonfatal overdose during prior 30 days: 17.7% vs. 7.0% (OR = 2.83, 95% CI 1.05–7.61, $p < 0.05$) |
| | | | Continuous engagement in MMT: 26.0% vs. 45.2% (OR = 0.43, 95% CI 0.21–0.88, $p < 0.05$) |
| Moore et al., 2018 [62] | Clinical trial: MMT vs. forced tapered withdrawal during incarceration | N = 382 | MMT group (during incarceration) more likely to start community-based MMT within 1 day post-release (OR = 32.04, 95% CI 7.55–136.01, $p < 0.001$) |
| | | | MMT group more likely to start community-based MMT within 30 days post-release (OR = 6.08, 95% CI 3.43–10.79, $p < 0.001$) |
| McMillan et al., 2008 [63] | Cohort with jail-based MMT | N = 589 | No statistically significant effect of jail-based MMT on re-incarceration (HR = 1.16, 95% CI 0.81–1.68) |
| | | | No statistically significant effect of MMT dosage received upon release on re-incarceration rate (HR = 1.05 per additional 10 mg, 95% CI 0.99–1.12). |
| | | | Data do not support the hypothesis that jail-based MMT increases or reduces re-incarceration |
| Marzo et al., 2009 [64] | Cohort: OAT (MMT or BPN) in prison | N = 507 (MMT, 104; BPN, 290; no OST, 113) | OAT delivered to 77.7% of opioid-dependent patients during imprisonment |
| | | | After adjustment for confounders, MMT not associated with a reduced rate of re-incarceration (aRR = 1.28, 95% CI 0.89–1.85, $p = 0.19$) |
| Wickersham et al., 2013 [65] | Cohort: methadone dose at release <80 mg/daily vs. ≥80 mg/daily | N = 27 | Methadone dose of ≥80 mg/daily associated with treatment retention |
| | | | At 12 months post-release, 21.4% of participants on <80 mg/daily were retained vs. 61.5% of those on ≥80 mg/daily ($p < 0.01$) |

*(Continued)*

**Table 3.** (Continued)

| Source | Intervention | Sample size | Outcomes/conclusions |
|---|---|---|---|
| Wickersham et al., 2013 [66] | Cohort: methadone dose at release <80 mg/daily vs. ≥80 mg/daily | N = 72 | Methadone dose of ≥80 mg/daily associated with treatment retention |
| | | | At 12 months post-release, 29.0% of participants on ≤80 mg/daily were retained vs. 73.1% of those on >80 mg/daily ($p < 0.001$) |
| Westerberg et al., 2016 [67] | Cohort: MMT in jail vs. opioid detox | N = 960 | Participants who received MMT in prison less likely to be re-incarcerated after 1 year vs. opioid detox group (53.4% vs. 72.2%, $p < 0.001$) |
| | | | Among those re-incarcerated within 1 year, the number of days to rebooking was longer for participants who received MMT in prison vs. opioid detox group (275.6 days [SD 124.9] vs. 236.3 days [SD 131.2], $p = 0.035$) |
| | | | 97.8% of participants who received MMT in prison continued MMT post-release |
| **Buprenorphine maintenance treatment** | | | |
| Sheard et al., 2009 [56] | RCT (2 groups): BPN vs. DHC in prison | N = 90 (BPN, 42; DHC, 48) | At 5 days post-detox: BPN group more likely to have a negative opioid test (RR = 1.61, 95% CI 1.02–2.56, $p = 0.04$) |
| | | | At 1, 3, and 6 months: no statistically significant differences found between the groups |
| Magura et al., 2009 [57] | RCT (2 groups): BPN vs. MMT in jail | N = 116 (BPN, 60; MMT, 56) | While incarcerated, BPN group more likely to report to assigned addiction treatment post-release vs. MMT (48% vs. 14%, $p < 0.001$) |
| | | | While incarcerated, BPN group more likely to report intention to continue addiction treatment post-release vs. MMT (93% vs. 44%, $p < 0.001$) |
| | | | BPN group less likely to withdraw voluntarily from medication while in jail vs. MMT (3% vs. 16%, $p < 0.05$) |
| Awgu et al., 2010 [58] | RCT (2 groups): BPN vs. MMT in jail | N = 114 (BPN, 60; MMT, 54) | MMT patients reported more side effects than BPN patients, including depression ($p < 0.01$), constipation ($p < 0.01$), confusion ($p < 0.01$), and fatigue/weakness ($p < 0.05$) |
| | | | MMT patients reported more opioid withdrawal symptoms (85% vs. 53%, $p < 0.001$) |
| Wright et al., 2011 [59] | RCT: MMT vs. BPN | N = 289 (MMT, 148; BPN, 141) | MMT vs. BPN had equal clinical effectiveness in achieving abstinence at follow-up |
| | | | Predictors of abstinence: continued incarceration and abstinence during previously measured period |
| | | | **Impact of incarceration vs. release on abstinence** |
| | | | At 8 days post-detoxification, participants still in prison were more likely to be abstinent than those released to the community: OR = 15.2 (95% CI 4.2–55.3, $p < 0.001$) |
| | | | At 1 month post-detoxification, participants still in prison were more likely to be abstinent than those released to the community: OR = 7.0 (95% CI 2.2–22.2, $p = 0.001$) |
| | | | **Impact of abstinence on previous measurement of current abstinence** |
| | | | At 1 month of follow-up, abstinence at 8 days of follow-up vs. not: OR = 4.5 (95% CI 1.9–10.3, $p < 0.001$) |
| | | | At 3 of months follow-up, abstinence at 1 month of follow-up vs. not: OR = 8.6 (95% CI 3.2–23.3, $p < 0.001$) |
| | | | At 6 months of follow-up, abstinence at 3 months of follow-up vs. not: OR = 32.8 (95% CI 6.1–176.6, $p < 0.001$) |

AHR, adjusted hazard ratio; aRR, adjusted relative risk; BPN, buprenorphine; DHC, dihydrocodeine; HR, hazard ratio; MMT, methadone maintenance treatment; OR, odds ratio; OST, opioid substitution treatment; OAT, opioid agonist treatment; PY, person-years; RR, relative risk.

Island (RIDOC) who had been engaged in MMT prior to their incarceration. Participants were randomly assigned into 1 of 2 groups, with the first group receiving continued access to MMT during their incarceration period ($n = 128$) and the second group undergoing tapered withdrawal from MMT within 1 week of their incarceration ($n = 51$). At 12 months post-

release, participants who were assigned to tapered withdrawal from methadone after the first week of incarceration were more likely to use heroin (OR = 2.02, 95% CI 1.01–4.04, $p < 0.05$) and injection drugs (OR = 2.95, 95% CI 1.43–6.06, $p < 0.05$) compared to those who continued MMT while incarcerated.

Multiple studies have found that individuals with OUD who receive MMT during their incarceration period are more likely to access, initiate, and adhere to addiction treatment post-release, compared with individuals who do not receive MMT during their incarceration [60–62]. Findings from the RCT conducted by Brinkley-Rubinstein et al. [61] also indicated that those who were assigned to tapered withdrawal from methadone after the first week of incarceration were significantly more likely to experience a nonfatal overdose during the 12-month follow-up period, compared to those who received MMT while incarcerated (OR = 2.83, 95% CI 1.05–7.61, $p < 0.05$). Rich et al. [60] also conducted an RCT with 223 incarcerated persons from Rhode Island who had been engaged in MMT at the time of their arrest, comparing a group of participants who were maintained on MMT while incarcerated ($n = 114$) with a second group who underwent forced tapered withdrawal from MMT within 1 week of incarceration ($n = 109$). Participants who were maintained on MMT while incarcerated were 2 times more likely to return to a community methadone clinic within 1 month of release compared to incarcerated persons who underwent forced tapered withdrawal (HR = 2.04, 95% CI 1.48–2.80, $p < 0.001$).

A trial conducted by Moore et al. [62] compared incarcerated persons who continued engagement in MMT ($n = 184$) with a control group who underwent forced tapered withdrawal from MMT while in jail ($n = 198$). The authors found that participants who were maintained on MMT during incarceration were more likely to engage in community-based MMT within 1 day (OR = 32.04, 95% CI 7.55–136.01, $p < 0.001$) and 30 days post-release (OR = 6.08, 95% CI 3.42–10.79, $p < 0.001$), compared to incarcerated persons with OUD who underwent forced tapered withdrawal from MMT.

Wickersham et al. [65,66] conducted a longitudinal study among 72 HIV-positive incarcerated persons with OUD opioid dependence in Malaysia. Findings indicated that receiving MMT during incarceration and receiving a dose of ≥80 mg/day at the time of release was significantly associated with retention in addiction treatment at 12 months post-release compared to receiving a dose of <80 mg/day at the time of release (73.1% versus 29.0%, $p < 0.001$).

Westerberg et al. [67] examined re-incarceration outcomes among 960 incarcerated persons from a large US metropolitan detention center. The study compared individuals with OUD who were enrolled in community MMT prior to incarceration and continued MMT during their incarceration period ($n = 118$) with 3 groups: incarcerated individuals with no known substance use disorders ($n = 385$), incarcerated individuals receiving alcohol detoxification ($n = 220$), and incarcerated individuals receiving opioid detoxification ($n = 237$). The authors found that those enrolled in MMT while incarcerated were less likely to be re-incarcerated within 1 year post-release, compared to those who received opioid detoxification while incarcerated (53.4% versus 72.2%, $p < 0.001$). Among individuals who were re-incarcerated within 1 year, the number of days to rebooking into the detention center was higher for participants who received MMT during incarceration compared to the opioid detoxification group (275.6 days versus 236.3 days, $p = 0.035$). Furthermore, among individuals who received MMT while incarcerated, 97.8% continued to be enrolled in MMT at community clinics post-release.

**Buprenorphine maintenance treatment.** Four studies compared the effects of providing BMT relative to other—primarily pharmacological—forms of treatment for OUD during incarceration [56–59]. Sheard et al. [56] conducted an RCT with 90 incarcerated persons from a large prison in England, comparing 1 group of participants who received daily sublingual BPN over 20 days ($n = 42$) with a second group receiving daily oral dihydrocodeine over 20

days ($n = 48$) on abstinence from illicit opiates. At 5 days post-detoxification, participants receiving BPN were more likely to present a negative urine test for opiates compared to those who received dihydrocodeine (57% versus 35%, relative risk = 1.61, 95% CI 1.02–2.56).

Magura et al. [57] conducted a study in 2009 among heroin-dependent incarcerated persons at Rikers Island in New York City ($n = 116$) to compare the effects of in-jail BMT and MMT. Participants were randomly assigned to receive BMT ($n = 60$) or MMT ($n = 56$). The authors found that incarcerated persons receiving BMT were less likely to withdraw from medication while in jail (3% versus 16%, $p < 0.05$) and more likely to report to assigned addiction treatment post-release (48% versus 14%, $p < 0.001$), compared to those receiving MMT while incarcerated. However, they observed no differences in self-reported rates of relapse into illicit opioid use post-release or in rates of re-incarceration between the 2 groups. Awgu et al. later [58] conducted a clinical trial in 2010 to compare in-jail BMT and MMT among 133 incarcerated persons with OUD at Rikers Island, with 77 participants enrolled in the BMT group and 56 participants enrolled in the MMT group. The authors found that participants who received in-jail MMT reported higher levels of opioid withdrawal symptoms (85% versus 53%, $p < 0.001$) and more side effects, including depression ($p < 0.01$), constipation ($p < 0.01$), confusion ($p < 0.01$), and fatigue/weakness ($p < 0.05$).

Wright et al. [59] compared the effects of receiving either a methadone or BPN detoxification during a reduced regimen of not more than 20 days among 306 UK incarcerated persons with OUD. Participants were randomly assigned to receive daily sublingual BPN ($n = 141$) or to receive oral methadone treatment ($n = 148$). Eight days after the detoxification, 73.7% of participants achieved abstinence—with no statistically significant difference between the BPN and MMT groups (OR = 1.69, 95% CI 0.81–3.51, $p = 0.163$). Incarceration was the major predictor of abstinence, compared to prison release (Table 3).

### Opioid use interventions post-incarceration

**Study characteristics.** Eleven studies evaluated opioid interventions delivered after incarceration, with follow-up periods ranging from 1 month to 7 years post-release (Table 4). Seven of these studies were observational [69–75], and 4 were experimental [76–79]. Two studies evaluated the impact of post-release MMT continuation on all-cause and overdose mortality [69] and on criminal recidivism [71]. Five studies evaluated the impact of BPN-based opioid pharmacotherapy post-release on access to and/or retention in addiction treatment, levels of illicit opioid use, and criminal recidivism [70,72,73,76,77]. One RCT compared the efficacy of MMT and naltrexone implants post-release in reducing illicit drug use and criminal recidivism [78], and 3 studies evaluated the impact of NLX kit provision in preventing overdose-related deaths post-release [74,75,79].

**Methadone maintenance treatment.** Two cohort studies evaluated the impact of providing MMT to incarcerated individuals post-release [69,71]. Huang et al. [69] analyzed administrative data from a cohort of 4,357 incarcerated persons with a history of opiate injection. The study found that 46% of individuals ($n = 1,982$) had enrolled in MMT by 18 months after their release from prison; however, the majority ($n = 1,282$, 65%) discontinued MMT shortly after enrolling. MRs were lowest among individuals who remained in MMT during the follow-up period (MR = 0.24/100 PY), followed by those who did not enroll in MMT post-release (MR = 2.6/100 PY). The highest mortality rate was observed among MMT enrollees who dropped out of treatment (MR = 7.0/100 person-years). Post-release MMT attendance had a significant protective effect on both overdose mortality (HR = 0.09, $p = 0.02$) and all-cause mortality (HR = 0.07, $p < 0.001$). Moreover, risk of re-incarceration was found to be the lowest among those who continued MMT post-release (3.4%), compared to those who had never

**Table 4.** Description of intervention and related outcomes of opioid use interventions post-incarceration, 2008–2019.

| Source | Intervention | Sample size | Outcomes/conclusions |
|---|---|---|---|
| **Methadone maintenance treatment** | | | |
| Huang et al., 2011 [69] | Cohort: MMT post-release | N = 4,357 (MMT, 1,982; no MMT, 2,375) | 46% of participants enrolled in MMT post-release, with 127 participants (6%) enrolled during the first 30 days post-release |
| | | | Significant protective effect of MMT attendance on all-cause mortality (HR = 0.07, 95% CI 0.02–0.21, $p < 0.001$). |
| | | | **Mortality rate (deaths per 100 PY)** |
| | | | During MMT (0.24 deaths per 100 PY, 95% CI 0.08–0.74) |
| | | | Not enrolled on MMT (2.6 deaths per 100 PY, 95% CI 2.1–3.1) |
| | | | After MMT drop-out in community (7.0 deaths per 100 PY, 95% CI 4.8–10.2) |
| Macswain et al., 2014 [71] | Cohort: MMT post-release—continued MMT (MMT-C), discontinued MMT (MMT-T), no MMT (MMT-N) | N = 856 (MMT-C, 161; MMT-T, 481; MMT-N, 214) | Incarcerated persons who continued MMT post-release had a 36% lower risk of recidivism vs. non-MMT-treated group (AHR = 0.64, 95% CI 0.47–0.88, $p < 0.01$) |
| | | | No significant difference between the MMT-T and MMT-N groups |
| **Buprenorphine maintenance treatment** | | | |
| Lee et al., 2012 [70] | Cohort: BPN in jail vs. community referral | N = 142 (jail, 32; community, 110) | Similar treatment retention and rates of opioid abstinence between jail-treated vs. community-referred patients |
| | | | **BPN treatment retention** |
| | | | At 48 weeks: 37% for BPN in jail vs. 30% for community referral |
| | | | Mean opioid use decreased from 7 days/week at prearrest/induction visit to 1 day/week at week 12 among overall sample, with no statistically significant difference between the 2 groups |
| Fox et al., 2014 [72] | Cohort: BPN/NLX post-release | N = 27 | At 1-month follow-up, 82% of participants were retained in care and 44% had reduced opioid use |
| | | | At 6-months follow-up, 33% of participants were retained in case and 19% had reduced opioid use |
| Riggins et al., 2017 [73] | Cohort: BMT during first 30 days post-release | N = 306 | **After adjustment for potential confounding variables, recent incarceration was not significantly associated with outcomes** |
| | | | BMT retention at 6 months (OR = 0.95, 95% CI 0.46–1.98) |
| | | | BMT retention at 12 months (OR = 0.57, 95% CI 0.27–1.18) |
| | | | Self-reported opioid use (OR = 0.99, 95% CI 0.51–1.92) |
| Springer et al., 2010 [76] | RCT (2 groups): BPN/NLX vs. MMT incarcerated persons recently released (<90 days) | N = 23 (BPN/NLX) | **Evaluation of BPN/NLX group at 12 weeks of follow-up** |
| | | | 91% of participants completed the induction period (approximately 3 days) |
| | | | Mean opioid craving (based on 10-point scale) decreased from 6 to 1.8 following induction; 2.2 at end of follow-up |
| | | | 74% retention rate after 12 weeks |
| | | | Positive urine test for opiates: 29% at baseline vs. 17% at follow-up |
| | | | Positive urine test for cocaine: 43% at baseline vs. 29% at follow-up (29%) |
| Springer et al., 2012 [77] | RCT: BPN/NLX vs. MMT among incarcerated persons recently released (<90 days) | N = 94 | The mean opioid craving score was 5.5 at the time of baseline induction, and reduced to 1.0 by the end of week 1 (opioid craving remained consistent at the 24-week end point) |
| | | | Satisfaction with BPN/NLX treatment was high, with a mean satisfaction score of 9 by the end of the first week of induction rising to a mean of 10 throughout the rest of the 24 weeks of the study (10-point scale) |
| **Naltrexone implant** | | | |
| Lobmaier et al., 2010 [78] | RCT: MMT vs. NTX implant (pre-release) | N = 44 (MMT, 21; NTX, 23) | NTX implant group more likely to be on medication after 6 months of follow-up vs. MMT group (69.6% vs. 23.8%, $p = 0.003$) |
| | | | There were no statistically significant differences between NTX and MMT groups in substance use and criminal activity |
| | | | Re-incarceration rates were comparable in both groups: 21.7% in NTX group and 23.8% in MMT group spent 1 or more days in a Norwegian prison during follow-up |

*(Continued)*

**Table 4.** (Continued)

| Source | Intervention | Sample size | Outcomes/conclusions |
|---|---|---|---|
| **Naloxone kit** | | | |
| Bird et al., 2016 [74] | Cohort: before vs. after start of program dispensing NLX kit upon release | N = 2,273 NLX kits issued to incarcerated persons upon prison release | **ORDs before vs. after start of Scotland's National Naloxone Program** |
| | | | 2006–2010: 9.8% ORDs (193 ORDs among people released from prison of 1,970 ORDs registered in the period) |
| | | | 2011–2013: 6.3% ORDs (76 ORDs among people released from prison of 1,212 ORDs registered in the period) |
| | | | Decrease of ORDs: 3.5% (95% CI 1.6%–5.4%, p < 0.001) |
| Bird et al., 2017 [75] | Cohort: NLX kit upon release | N = 4,124 NLX kits issued to incarcerated persons upon prison release | **Observed ORDs within 4 weeks of prison release** |
| | | | 2006–2010: 193 ORDs among people released from prison of 1,970 ORDs registered in the period (9.8%, 95% CI 8.5%–11.1%) |
| | | | 2011–2012: 76 ORDs among people released from prison, of 1,212 ORDs registered in the period (6.3%, 95% CI 4.9%–7.6%) |
| | | | 2014–2015: 37 ORDs among people released from prison, of 942 ORDs registered in the period (3.9%, 95% CI 2.7%–5.2%) |
| | | | 60% reduction in ORDs observed in the 2014–2015 calendar year vs. 2006–2010 |
| Parmar et al., 2017 [79] | RCT: NLX kit upon release vs. no overdose prevention kit | N = 1,557 | 842 NLX kits delivered and compared to 843 control (empty) kits |
| | | | Participants who carried NLX kit (or empty kit) during the first 4 weeks post-release: 75% (95% CI 63%–79%) |
| | | | Participants present during an opioid-related overdose: 80% (95% CI 75%–84%) |
| | | | Provision of NLX upon release is feasible, is acceptable, and may be life-saving to prevent opioid-related overdose |

AHR, adjusted hazard ratio; BMT, buprenorphine maintenance treatment; BPN, buprenorphine; HR, hazard ratio; MMT, methadone maintenance treatment; NLX, naloxone; NTX, naltrexone; OR, odds ratio; ORD, opioid-related death; PY, person years.

enrolled in MMT (40.0%) or dropped out of MMT during the follow-up period (61.1%, p < 0.001).

Macswain et al. [71] examined data from a cohort of 856 incarcerated persons with OUD at a Canadian federal correctional facility, to determine the impact of MMT on re-incarceration. The study compared 3 groups: individuals who received MMT while incarcerated and continued treatment in the community post-release (n = 161), individuals who terminated MMT involvement upon release (n = 481), and individuals who did not participate in MMT during their incarceration (n = 214). The authors found that individuals who continued engagement in MMT post-release had a 36% lower risk of criminal recidivism, compared to those who did not engage in MMT during their incarceration (AHR = 0.64, 95% CI 0.47–0.88, p < 0.01). No significant difference in recidivism risk was found between those who terminated MMT involvement upon release and those who did not engage in MMT during their incarceration.

**Buprenorphine maintenance treatment.** Five studies evaluated the impact of post-release BPN-based treatments on treatment retention, illicit opioid use, and criminal recidivism [70,72,73,76,77]. Springer et al. [76,77] examined data from a group of HIV-positive incarcerated persons with OUD who received BPN/NLX treatment post-release. The authors found high levels of treatment retention, with 91% of participants completing BPN/NLX induction and 74% remaining on treatment after 12 weeks. Participants further reported reduced opioid cravings and consistently high levels of treatment satisfaction at 12 weeks follow-up, and adverse events were found to be mild and few.

A study conducted by Lee et al. [70] compared treatment retention and opioid misuse among opioid-dependent adults seeking BPN/NLX treatment in the community, including one group of incarcerated persons recently released (*n* = 32, 22 of whom initiated BPN/NLX while incarcerated) and a second group of patients with no recent incarceration history (*n* = 110). At 48 weeks, the authors found similar BPN/NLX retention rates among previously incarcerated persons (37%) and general patients (30%). Positive urine test results for opiates, as well as rates of self-reported opioid misuse, were also similar between the 2 groups.

Fox et al. [72] examined electronic medical record data to determine health outcomes and healthcare utilization among a group of opioid-dependent patients (*n* = 27) released from the New York State Department of Corrections between 2009 and 2013. At 1-month follow-up, 82% of individuals were retained in BPN/NLX treatment and 44% had reduced opioid use (defined as ≥50% of urine drug test results without opioids). After 6 months of follow-up, 33% of participants were retained on BPN/NLX treatment, and 19% had reduced their opioid use.

Riggins et al. [73] analyzed data from a cohort of HIV-positive patients diagnosed with opioid dependence who were receiving BMT. Among patients incarcerated in the previous 30 days, those who were retained in BMT had lower rates of subsequent incarceration at 6 months (8.4% versus 27.9%, *p* < 0.01), 9 months (4.9% versus 20.4%, *p* < 0.01), and 12 months of follow-up (6.2% versus 24.8%, *p* < 0.01), compared to those who were not retained in BMT.

**Naltrexone implants.** Lobmaier et al. [78] conducted an RCT with 44 incarcerated persons with OUD in Norway comparing treatment outcomes and retention among 2 groups: individuals who received naltrexone implants 1 month prior to release (*n* = 23) and individuals who received MMT 1 month prior to release (*n* = 21). After 6 months of follow-up, relapse to heroin use was less likely among the naltrexone group relative to the MMT group (*p* = 0.012). Moreover, at 6 months follow-up post-release, 69.6% of the participants in the naltrexone implant group remained engaged in the naltrexone treatment, compared to 23.8% in the MMT group (*p* = 0.003). Rates of re-incarceration during the 6-month follow-up period were comparable across both groups, at 21.7% in the naltrexone group and 23.8% in the MMT group.

**Naloxone kit provision.** Three studies evaluated the impact of providing NLX kits to incarcerated persons upon release from a correctional institution, as a strategy to prevent opioid-related overdose deaths [74,75,79]. Parmar et al. [79] conducted a large-scale RCT among UK incarcerated persons with history of heroin injection (*n* = 1,557) in order to evaluate the feasibility of providing NLX kits upon prison release to prevent opioid-related overdose. Participants were randomly assigned to either the experimental group, where they received a pack containing a single "rescue" injection of NLX (*n* = 842), or the control group, where they received a pack containing a placebo injection (*n* = 843). Randomization was prematurely discontinued as it was found that only one-third of NLX administrations were delivered to former incarcerated persons. The study ultimately included results from 205 former incarcerated persons, with 112 in the experimental group and 93 in the control group. Twenty-one percent of experimental group participants reported administration of NLX to themselves or to someone else before the arrival of a doctor or ambulance, compared to 9% for the control group (*p* = 0.02).

Bird et al. conducted 2 studies [74,75] to evaluate the impact of Scotland's implementation of the National Naloxone Program (NNP) for reducing opioid-related deaths, which involved training about NLX administration and distribution of NLX kits to at-risk incarcerated persons upon prison release. The first study assessed data from 2,273 formerly incarcerated persons, and found that post-release opioid-related deaths decreased significantly following the implementation of the program [74]. Between 2006 and 2010—prior to the introduction of the NNP—9.8% of opioid-related deaths were among formerly incarcerated persons within 4 weeks of release (*p* < 0.001). Between 2011 and 2013—after NNP implementation—6.3% of

opioid-related deaths were among formerly incarcerated persons within 4 weeks of release ($p < 0.001$). Bird et al. conducted a subsequent analysis arriving at similar results: specifically, the authors reported a 60% reduction in the proportion of opioid-related deaths within 4 weeks of prison release from 2006–2010 (i.e., prior to program implementation) to 2014–2015 [75]. Rates of opioid-related deaths within 4 weeks of prison release were 19.8% in 2006–2010, 6.3% in 2011–2013, and 3.9% in 2014–2015, illustrating a continuous decrease in overdose-related deaths among former Scottish incarcerated persons, largely attributed to the introduction of the NNP.

## Discussion

The objective of this systematic review was to describe the impact of opioid-related interventions delivered during and after incarceration among adult correctional populations. A strong evidence base exists emphasizing the positive impact of providing opioid-related interventions to incarcerated persons with OUD, particularly during a continuum of treatment prior to, during, and after incarceration. A key finding in this review is that pharmacological interventions including MMT, BPN/NLX, and NLX have positive impacts on post-release mortality, substance use, treatment adherence, and criminogenic outcomes if treatment is administered during incarceration and continued upon release [35,41,48,52,53,71]. Evidence from this review also suggests that incarcerated individuals who are exposed to OAT in correctional institutions are more likely to be engaged and retained in community-based treatments upon release.

Studies identified the effectiveness of providing opioid-related interventions during a continuum of treatment (before, during, and/or after incarceration), in support of the highly important concept of the "criminal justice continuum of care" for opioid users at risk of overdose, as introduced by Dr. Lauren Brinkley-Rubinstein and colleagues [13]. Our systematic review identified that receiving OAT during incarceration [35,36,51,53,54] was associated with reduced risk of re-incarceration [51], higher treatment adherence [53], and reduced mortality risk, both during and after incarceration [35,36,54]. Similarly, receiving MMT was, in general, associated with higher community-based treatment adherence and retention, reduced illicit opioid use, and reduced re-incarceration [38,39,43–45,52]. Receiving BPN while incarcerated [37,40,47] was also associated with higher treatment adherence post-release and lower rates of substance use and re-incarceration, compared with starting BPN treatment post-release [47]. In the same regard, XR-NTX treatment during incarceration [41,48–50,55] was associated with reduced relapse into illicit opioid use [41,48,49], reduced re-arrest likelihood [41], and fewer hospital admissions [50], along with higher treatment retention [55] and reduced overdose events, compared to receiving other treatments, including BPN or MMT [48]. However, the impact of XR-NTX treatment on re-incarceration rates remains unclear and warrants further research and evaluation.

There is also strong evidence from studies analyzing the impact of opioid-related interventions during incarceration. Receiving OAT during incarceration was associated with reduced opioid-related overdose deaths post-incarceration [26,68] and increased community-based treatment entry [68]. Similarly, receiving MMT was associated with better health outcomes [61], including reduced nonfatal overdoses [62] and increased treatment adherence and retention [60,62,65,66,67]. Compared to incarcerated individuals undergoing tapered withdrawal from MMT, those who continue receiving MMT while in correctional institutions have reduced illicit opioid use [61] and higher rates of return to community-based treatment upon release [60]. Compared to incarcerated individuals who received opioid detoxification, individuals who received MMT were less likely to be re-incarcerated [67]. Similar findings were identified among individuals receiving BPN in correctional institutions [56–59], including

reduced opioid use [56], increased treatment adherence [57], and lower withdrawal symptoms/treatment side effects, compared to those receiving MMT [58].

Studies evaluating the impact of opioid-related interventions post-incarceration identified that continuation of MMT decreases the risk of both overdose mortality [69] and re-incarceration [69,71]. Post-incarceration interventions involving BPN treatment also increase addiction treatment adherence, decrease opioid craving, and consequently reduce opioid use [70,72,73,76]. Studies evaluating the impact of naltrexone implants post-incarceration should be conducted with larger sample sizes.

Overall, the review suggests positive effects of pharmacological interventions in mitigating harmful effects of problematic opioid use among the correctional population [35,41,43,48,52,71]. The results are also consistent with observations in evaluations of opioid-related interventions provided to the general population, especially regarding reduced opioid-related mortality. Nonetheless, comparisons of intervention outcomes between these different populations are limited, considering that accessibility to services is more restricted in the correctional population, along with other program delivery and implementation barriers.

In any stage of the criminal justice continuum in which OAT interventions were provided, they were associated to reduced mortality risks [26,35,36,54,68] and higher treatment adherence [53,68]. In the same regard, MMT was found to also increase treatment adherence and retention [38,39,43–45,52,60–62,65–67], reduce re-incarceration [52,60,67,69,71], and decrease overdose mortality when continued post-release [69]. Receiving BPN was found to increase OUD treatment adherence and decrease opioid use [37,40,47,57,70,72,73,76].

Our systematic review, thus, reinforces the positive impact of providing OAT in correctional settings. OAT decreases mortality rates, reduces opioid use, and improves addiction treatment intake and retention post-incarceration. Moreover, interventions such as immediate linkage with OAT services and the provision of preventive interventions (e.g., NLX kits) upon release are effective in decreasing overdose mortality in the often precarious weeks following incarcerated individuals' release. Findings from this review highlight the positive impact that re-entry programs may have on incarcerated individuals who are transitioning back to the community, especially since the post-release period is characterized by poor continuity of care, inadequate social support, and increased rates of opioid-related overdose and mortality [12,13].

Previous studies have shown that the period immediately following release from jail or prison is a critical time point, during which opioid-related overdose deaths are highly prevalent, often as the result of lowered opioid tolerance developed during the incarceration period [5,6,12]. Our review determined that providing NLX kits to incarcerated individuals upon release from a correctional institution may aid in reducing opioid-related deaths post-release [74,75,79].

There is, however, significant variation across countries related to the criminal justice system, legislation towards addiction, and the healthcare system. While some countries, such as Canada [52] and Australia [36], offer OAT to incarcerated persons with OUD, in the US, prisons and jails frequently adopt an abstinence-only approach [12]. This is of great concern, since the US has the highest incarceration rate in the world, with over 2 million incarcerated persons in jails and prisons [80]. Despite the strong evidence of improved health, behavioral, and social outcomes, in 2019 the majority of US jails and prisons did not offer OAT to individuals with diagnosed OUD. According to Dr. Joanne Csete [12], "Expanding access to treatment of opioid use disorder (OUD) is central to addressing the US overdose mortality crisis."

Our study corroborates this statement, identifying that incarcerated persons with OUD benefit from OAT, reaching similar levels of treatment adherence, health and social improvements as persons with OUD without incarceration history. Many persons with OUD

will continue to be involved in the criminal justice system [13] given the high rate of addiction relapse, recidivism, and re-incarceration [12]. Legislators, policy makers, public health professionals, researchers, and criminal justice representatives need to work together to address the dual crisis of mass incarceration and opioid-related overdose. The lack of effective treatment and opioid overdose prevention strategies for incarcerated persons with OUD is referred to, by some experts, as an unacceptable human rights violation [81], representing a key medical and public health concern [12,13,82].

This review has several limitations. First, we did not assess for any risk of biases, as the included studies were heterogeneous in terms of study design and settings, participant characteristics (e.g., type of opioid used, history of opioid use, length of incarceration, time of release from correctional institution), methods of data analysis, and reporting of findings. Receiving or initiating OAT in short-term (jails) versus long-term (prisons) incarceration facilities might affect OAT effectiveness in ways that are beyond the scope of this systematic review, and the impact of OAT provided in these different settings should be further evaluated. Second, the international scope of the review limited the generalizability of results, considering the potential influence of country-specific contexts, such as varied criminal justice systems, drug laws, correctional policies, and healthcare access. Third, the determination of whether there was a meaningful effect for each study outcome was based on statistical significance, which does not necessarily represent clinical or population-level significance. Fourth, considering the differences in program delivery, we excluded any study that focused on individuals involved in other areas of the criminal justice system (e.g., probationers, parolees), which excluded a large number of studies that assessed the effects of treatment and prevention interventions for other criminal-justice-involved individuals with OUD. Moreover, this review does not contain any grey literature; thus, any relevant studies that may been conducted but are not published have been excluded from the results.

In spite of the above limitations, our review reinforces the key benefits of providing opioid-related interventions within correctional settings. Our study also highlights the need to implement and scale up evidence-based strategies to ensure incarcerated individuals with OUD are able to access adequate treatment and care during and post-incarceration. Inadequate linkage to and provision of care for problematic opioid use and OUD immediately post-incarceration continues to serve as a missed opportunity to address the high rates of overdose and mortality typically observed among incarcerated adults living with OUD within the first weeks following release. The delivery of opioid-related interventions in correctional institutions constitutes a unique opportunity to provide treatment for a high-risk sub-group of opioid users, thus contributing to reduced illicit opioid use and high-risk behaviors within correctional settings and in the community.

## Supporting information

**S1 List of Manuscripts Excluded. List of full-text papers evaluated and excluded.**
(DOCX)

**S1 MEDLINE Search Strategy.**
(DOCX)

**S1 PRISMA Checklist.**
(DOC)

**S1 Prospero Registration.**
(PDF)

## Author Contributions

**Conceptualization:** Monica Malta, Benedikt Fischer.

**Data curation:** Monica Malta, Sarah Bonato.

**Formal analysis:** Monica Malta, Thepikaa Varatharajan, Cayley Russell, Michelle Pang.

**Funding acquisition:** Benedikt Fischer.

**Methodology:** Monica Malta, Thepikaa Varatharajan, Cayley Russell, Sarah Bonato.

**Project administration:** Monica Malta, Thepikaa Varatharajan, Cayley Russell, Michelle Pang, Benedikt Fischer.

**Supervision:** Monica Malta, Benedikt Fischer.

**Writing – original draft:** Monica Malta, Thepikaa Varatharajan, Cayley Russell, Michelle Pang.

**Writing – review & editing:** Monica Malta, Thepikaa Varatharajan, Cayley Russell, Michelle Pang, Sarah Bonato, Benedikt Fischer.

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
