## [Decision Letter · Decision Letter 0]

1 Oct 2019

Dear Dr. Malta,

Thank you very much for submitting your manuscript "Opioid-Related Treatment, Interventions and Outcomes among Correctional Populations: A Systematic Review" (PMEDICINE-D-19-02187) for consideration at PLOS Medicine. 

[LINK]

In light of these reviews, I am afraid that we will not be able to accept the manuscript for publication in the journal in its current form, but we would like to consider a revised version that addresses the reviewers' and editors' comments. Obviously we cannot make any decision about publication until we have seen the revised manuscript and your response, and we plan to seek re-review by one or more of the reviewers. 

We expect to receive your revised manuscript by Oct 15 2019 11:59PM. Please email us (plosmedicine@plos.org) if you have any questions or concerns.

We look forward to receiving your revised manuscript. 

Sincerely,

Clare Stone PhD

Acting Chief Editor 

for 

Richard Turner, PhD

Senior Editor 

PLOS Medicine

plosmedicine.org

AE comments:

1. Please include in an Appendix document a list of the 38 studies deemed ineligible for reasons of aim/outcome/intervention focus.

2. The first paragraph provides a reasonable summary of the evidence about health among incarcerated persons. But at least one additional point of the manuscript (or a point that the manuscript _should_ make) is that people with OUDs who have recently been incarcerated are a uniquely vulnerable population. Detoxification without linkage to OAT increases overdose risk substantially (Merrall EL et al, Addiction. 2010;105(9):1545-54. Binswanger IA et al, Ann Intern Med. 2013;159:592-600), but neither OAT nor linkage are commonly provided (Aronowitz SV et al, J Correct Health Care. 2016;22(2):98-108. Friedmann PD et al, Subst Abus. 2012;33(1):9-18).

3. At some point in the introduction or discussion, the authors should make the point that OAT has been found to reduce mortality in non-criminal justice involved populations (eg., Larochelle MR, Ann Intern Med. 2018;169(3):137-45. Sordo L et al, BMJ. 2017;357:j1550. Ma J et al, Mol Psychiatry, in press, doi:10.1038/s41380-018-0094-5). Benefits on a wide range of other non-mortality outcomes have also been described, including HIV transmission risk behavior. The discussion in particular would be enhanced if the authors contrasted their findings with studies of non-criminal justice involved populations. It is well known that OAT reduces mortality. It should not come as a surprise that OAT reduces mortality among those who have been incarcerated or recently released. ** It would be useful if the authors could contextualize their findings in light of this other literature-- are the effect sizes observed among incarcerated/released persons larger or smaller than in non-incarcerated populations (and why)? **

Please note that I am _not_ requiring that the authors cite the specific manuscripts listed above. Many are big papers in the field, but if the authors have alternative citations to support these claims, those could certainly be defended.

4. R3 requests that the phrase "during incarceration" be changed, but in my opinion it appears that there are several instances in the manuscript where the phrase "during incarceration" is appropriate and does not need to be changed.

5. I am in agreement with R3 that the Discussion--particularly paragraphs 2-4--is better described as a rehashing of the Results rather than a summative synthesis of the findings. Please rewrite as appropriate, synthesizing the findings across studies/across interventions/across timepoints in the criminal justice system cascade.

Editorial comments: 

Title – please provide country setting Abstract – related to above point, you open the abstract by mentioning the US specifically. Please state if the studies you select are based on US cohorts. I see some are from NSW, so perhaps the opening of the abstract could have a more global perspective? 

Abstract – please avoid a list like approach using numbers to list points and please provide a sentence of limitations as the final sentence of the Methods and Findings section of the abstract. Finally please avoid using causal language as this is a systematic review (strong evidence)

There is perhaps some kind of formatting issue in the methods section ((i.e. exp Opiate Substitution Treatment/, buprenorphine$, methadone$, naltrexone$, naloxone$) among incarcerated and post-incarceration populations (i.e. jail$, prison$, offender$, probat$, felon$, detention$, imprison$, postincarcerat$) who use opioids (i.e. exp Opioid-Related Disorders/, opiate$, opioid$)) – please correct here and elsewhere, as needed. 

Please provide 95% Cis as well as p values for all quantifiable data.

Please use the "Vancouver" style for reference formatting, and see our website for other reference guidelines https://journals.plos.org/plosmedicine/s/submission-guidelines#loc-references

Thank you for providing the PRISMA checklist – please use sections and paragraphs instead of page numbers, as these can change on revisions or publication. 

Comments from the reviewers:

Reviewer #1: See attachment

Michael Dewey

Reviewer #2: This is a systematic review of peer reviewed literature in the past decade (2008-2018) of opioid use interventions delivered during and post incarceration/release among adult correctional populations. The 45 selected studies from the 2,323 articles reviewed provide strong support for the positive impact of opioid related interventions during and post release from incarceration. The selections of the studies and the descriptions of the studies are appropriate, as are the conclusions.

It is important to note that incarceration and criminal justice involvement, in many countries, especially in the US, is part of the natural history of this disease. The introduction describes the "correctional population" as if they are a different species almost, and clearly there are differences, but it is important to keep in mind that this is the same disease.

It would be helpful to include in one of the tables whether the study was done in a jail, a prison population or combined, or other population. There should also be some brief discussion of the differences between interventions in jails (much larger numbers and shorter times incarcerated) and prisons (smaller numbers, more time to work with people).

p-5-6, sentence about linkage points should include probation and parole (as part of the "etc").

p-8, why did the authors decide to exclude drug courts and probation and parole? I agree with that choice but that should be described. [also, the authors should consider doing 2 more systematic reviews- one on drug courts and one on probation and parole- those would also be substantial contributions to the field]

p-8, line 4, when "initiated" is used, does that include people who were continued on medications during incarceration? I believe it does or should, and just needs to be explicit.

p-8, lines 8-10, should also explain why all of these studies are excluded

p-32, line 19 uses the term "offenders". Although that term is frequently used, it should be avoided as 1) it is further stigmatizing of an already severely stigmatized population, and 2) it is not always accurate, as many people are erroneously incarcerated even though they have not committed an offense.

Person first language is better. IE an incarcerated person.

p-37 line 16 typo "to into"

p-42, line 5, avoid the term "offenders"

p-42, line 10, what is "criminal recidivism"? is that re-incarceration? re-incarceration based upon new charges, a new conviction, or does it include also people incarcerated for parole or probation violations?

p-44, lines 5, 7, and 8, these percentages (and probably most in the article) should be rounded to whole numbers. The decimal points imply a degree of accuracy that the data certainly does not support and they are distracting.

Page 48, line 2- again, I would suggest a seperate paper reviewing treatment of opioid use disorder in probation and parole because the numbers of individuals is much larger.

p-48, line 13, "post-continues" is a typo

Reviewer #3: This is an important paper on a very important topic. However, the paper needs to be strengthened before it is ready for publication:

1) the terms used throughout are not consistently used or defined. A few examples: a) the current paradigm we are in is an overdose crisis, not an opioid crisis. Would change language throughout to emphasize this; b) suggest using person first language throughout rather than inmate, etc.; c) Often the authors talk about opioid use "during incarceration", but they don't mean this--they mean "among people with criminal justice experience. This should be changed throughout; d) they also often refer to "opioid use interventions", but this is not comprehensive enough. They mean opioid-related interventions more broadly because many interventions (for example, naloxone distribution) aren't targeting opioid use, but instead preventing overdose; e) finally, suggest using one term to refer to the criminal justice system throughout rather than incarceration, corrections, etc. 

2)In the methods they ought to explain why they looked at the years 2008-2018 and why they included only folks within 90 days of release.

3) On page 5 when talking about the criminal justice system continuum, they should refer to and/or be familiar with Brinkley-Rubinstein (2018) Criminal Justice Continuum for Opioid Users at Risk of Overdose: https://www.ncbi.nlm.nih.gov/pubmed/29544869

4) When describing the particular studies in the result it is important to be very precise. For example, in the implant naltrexone section, the authors refer to how those with the implant were more likely to uptake treatment in the community, but it is unclear at what time point (one month, six months??) and whether they mean naltrexone specifically or any MAT. This precision should be applied to all text when referring to specific studies. 

5) The discussion, in many ways, rehashes the results instead of expanding them and making recommendations for how they might be used to expand MAT program implementation. Suggest, adding more in this vein.

6) Finally, perhaps the biggest flaw of this paper, is the fact that there is no attention paid to country-specific context. For example the implantable naltrexone study (which there's only one of by the way) comes out of Norway, which has both a very, very different criminal justice landscape than the US and also more affordable access to healthcare. So, these results cannot be broadly applied to the US context--the authors must take the US context into consideration especially since the US overdose epidemic is the motivation for the paper in the first place. Therefore, for this paper to be acceptable for publication there must be some contextual overlay of the US criminal justice system and the lack of access to MAT in both corrections-based and community-based settings in the US.

[LINK]

---

## [Editor Report · Decision Letter 1]

4 Nov 2019

Dear Dr. Malta,

Thank you very much for re-submitting your manuscript "Opioid-Related Treatment, Interventions and Outcomes among Incarcerated Persons: A Systematic Review" (PMEDICINE-D-19-02187R1) for consideration at PLOS Medicine for our upcoming special issue on substance mis/use.

I have discussed the paper with editorial colleagues and our academic editor. I am pleased to tell you that, provided the remaining editorial and production issues are dealt with, we expect to be able to accept the paper for publication in the journal.

[LINK]

Please let me know if you have any questions. Otherwise, we look forward to receiving your revised manuscript shortly. 

Kind regards,

Richard Turner, PhD

rturner@plos.org

Requests from Editors:

You mention "prisons and jails" in your abstract, which might lead readers to wonder about potential differences between the two (also relevant to table 1). To avoid this, you could use a different term in the abstract, such as "correctional facilities", and perhaps add a sentence to your methods or results section to summarize the types of facility included in the studies and possibly relevant differences between them. 

To the abstract, please add some additional details summarizing the countries or regions in which the constituent studies were done; the number or proportions of those with randomized and observational research designs, respectively; the study sizes or aggregate numbers of participants; and breakdowns of the participants by sex. 

Please make that "fewer non-fatal overdoses" in the abstract; and quote the months of the start and end of the search period. 

We ask you to restructure your abstract so that the sentence summarizing study limitations is the final sentence of the "methods and findings" subsection. We suggest that lack of information about incarcerated populations that have not been the subject of published research could be quoted as a possible limitation.

The final two to three sentences of the "methods and findings" subsection of your abstract would appear to fit better into the "conclusion" subsection, and we ask you to move this text accordingly. Please begin the "conclusion" subsection with "In this study, we found that ..." or similar, and restrict this to around three sentences. 

Please refer to the attached Prospero and PRISMA documents early in your methods section. 

In the final sentence of the first paragraph of your discussion section, please adapt the wording to "be engaged and retained", or similar. 

On p.57 of the PDF, please begin a new paragraph quoting the study's limitations. 

Throughout your reference list, please abbreviate journal names consistently (e.g., "N Engl J Med" for reference 15). Please also add commas after the final author name preceding "et al." where needed, e.g., for reference 20.

Please add additional information for reference 32 as available. 

Please adapt your attached PRISMA checklist so that individual items are referred to by section (e.g., "Methods") and paragraph number rather than by page or line numbers, as the latter generally change in the event of publication. 

***

[LINK]

---

## [Editor Report · Decision Letter 2]

22 Nov 2019

Dear Dr. Malta, 

On behalf of my colleagues and the academic editor, Dr. Alexander C. Tsai, I am delighted to inform you that your manuscript entitled "Opioid-Related Treatment, Interventions and Outcomes among Incarcerated Persons: A Systematic Review" (PMEDICINE-D-19-02187R2) has been accepted for publication in PLOS Medicine. 

PRODUCTION PROCESS

PRESS

PROFILE INFORMATION

Thank you again for submitting the manuscript to PLOS Medicine. We look forward to publishing it. 

Best wishes, 

Richard Turner, PhD

Senior Editor 

PLOS Medicine

plosmedicine.org